# Bioinformatic Approach to Identify Positive Prognostic *TGFB2*-Dependent and Negative Prognostic *TGFB2*-Independent Biomarkers for Breast Cancers

**DOI:** 10.3390/ijms262311580

**Published:** 2025-11-29

**Authors:** Sanjive Qazi, Stephen Richardson, Mike Potts, Scott Myers, Vuong Trieu

**Affiliations:** 1Oncotelic Therapeutics, 29397 Agoura Road, Suite 107, Agoura Hills, CA 91301, USA; stephen.richardson@oncotelic.com (S.R.); michael.potts@oncotelic.com (M.P.); scott.myers@oncotelic.com (S.M.); vtrieu@oncotelic.com (V.T.); 2Westmorland Campus, Kendal College, Market Place, Kendal, Cumbria LA9 4TN, UK

**Keywords:** biomarker, breast cancer, prognosis, transforming growth factors, tumor microenvironment, immune

## Abstract

Breast cancer is highly heterogeneous, with multiple subtypes that differ in molecular and clinical characteristics. It remains the most common cancer among women worldwide. We conducted a hypothesis-generating study using a bioinformatics approach in order to identify potential prognostic biomarkers for breast cancer patients across multiple molecular subtypes. Given the influential role of the transforming growth factor beta (TGFB) pathway in shaping the immune microenvironment, we focused on the isoform, transforming growth factor beta 2 (*TGFB2*), which is upregulated in tumors, to identify *TGFB2*-dependent and -independent biomarkers for breast cancer patients’ overall survival (OS) responses. We evaluated the impact of *TGFB2* mRNA expression, in conjunction with other potential prognostic markers, on overall survival (OS) in breast cancer patients using The Cancer Genome Atlas (TCGA) and KMplotter databases. We employed a multivariate Cox proportional hazards model to compute hazard ratios (HRs) for *TGFB2* mRNA expression, integrating an interaction term that accounts for the multiplicative relationship between *TGFB2* and marker gene expressions while controlling age at diagnosis and cancer subtype and differentiating between patients receiving chemotherapy alone and those undergoing alternative therapeutic interventions. We used the KMplotter database to confirm *TGFB2*-independent prognostic markers from TCGA data. In cases dependent on *TGFB2*, increased mRNA expression of *TGFB2* alongside higher levels of *GDAP1*, *TBL1XR1*, *RNFT1*, *HACL1*, *SLC27A2*, *NLE1*, or *TXNDC16* was correlated with improved OS among breast cancer patients, of which four genes were upregulated in tumor tissues (*SLC27A2*, *TXNDC16*, *TBL1XR1*, *GDAP1*). Future studies will be required to confirm breast cancer patients could improve OS outcomes for patients expressing high levels of *TGFB2* and the marker genes in prospective clinical trials. Additionally, multivariate analysis revealed that the elevated expression of six genes (*ENO1*, *GLRX2*, *PLOD1*, *PRDX4*, *TAGLN2*, *TMED9*) were correlated with increases in HR, independent of *TGFB2* mRNA expression; all except *GLRX2* were identified as druggable targets. Future investigations assessing protein expression in breast cancer tumors to confirm the results of our retrospective analysis of mRNA levels will determine whether the protein products of these genes represent viable therapeutic targets. Protein–protein interaction (STRING) analysis indicated that TGFB2 is associated with EGFR and MYC from the PAM50 breast cancer gene signature. These findings suggest that correlation of *TGFB2*-related markers could potentially complement the PAM50 signature in the assessment of OS prognosis in breast cancer patients, but further validation of the TGFB2/EGFR/MYC proteins in tumors is warranted.

## 1. Introduction

Breast cancer is characterized as highly heterogeneous, comprising various subtypes that exhibit distinct molecular and clinical features. The incidence of breast cancer has been notably high among women globally, making it the most common type of cancer in this demographic. Research has identified bimodal age distribution patterns at diagnosis, with peaks around ages 45 and 65, suggesting the existence of two main etiological subtypes influenced by shared risk factors [1]. Characterization of breast cancer subtypes has evolved using molecular subtyping techniques, notably the PAM50 classification, which categorizes breast cancers into intrinsic subtypes, including luminal A, luminal B, HER2+, and triple-negative breast cancer (TNBC). Each subtype is associated with differing prognoses and therapeutic responses. For example, the luminal A subtype is often linked to better outcomes, while TNBC typically exhibits a higher mutational burden and poorer prognosis [2,3,4]. Additionally, recent studies have highlighted the role of immune profiles and biological processes in further stratifying these subtypes. For instance, a high-risk luminal A subtype has been identified with increased motility and immune dysfunction, correlating with poor prognosis [2]. Furthermore, the Breast Cancer Consensus Subtypes (BCCS) have classified tumors based on gene expression, revealing that estrogen receptor-positive (ER+) subtypes like *PCS1* and *PCS4* are associated with better prognoses, whereas subtypes like *PCS2* and *PCS3* are linked to poor outcomes [3].

We focused on the orchestrating role of the family of signaling molecules known as transforming growth factor beta (TGFB) to identify potential overall survival (OS) prognostic biomarkers in cancer patients. TGFBs regulate cell differentiation, proliferation, and selectivity through various signaling pathways. The TGFB signaling pathway plays a complex, two-phase role in cancer development. In the early stages, TGFB exhibits suppressive effects by inducing apoptosis and inhibiting cell cycle progression, thereby reducing the proliferation of cancer cells. However, as the disease progresses, cancer cells develop resistance to TGFB signals. Instead of inhibiting tumor growth, TGFB initiates cancer-promoting processes, including invasion, immunosuppression, angiogenesis, and the formation of metastasis [5,6,7,8,9]. Our previous studies identified the specific role of the transforming growth factor beta 2 (*TGFB2*) mRNA isoform as a negative prognostic marker in pediatric brainstem diffuse midline glioma tumors. They showed that high *TGFB2* mRNA levels were associated with an immunologically “cold” tumor microenvironment (TME), characterized by low expression of antigen-presenting cell (APC) markers (e.g., *CD14*, *CD163*, *ITGAX*/*CD11c*), which impairs anti-tumor immune responses [10]. In pancreatic ductal adenocarcinoma (PDAC), significant alterations in gene expression patterns were observed, particularly with a marked upregulation of *TGFB2* mRNA, which demonstrates a 7.9-fold increase over normal pancreatic tissue [11]. Additionally, interferon pathway genes, such as *IFNAR1*, *IRF9*, *STAT1*, and *IFI27*, were significantly upregulated, with *IFI27* exhibiting a highly significant 66.3-fold increase compared with normal tissue [11]. Notably, within TME characterized by low macrophage infiltration, high *TGFB2* mRNA expression proves particularly detrimental to survival, highlighting a median overall survival of 15.3 months compared with 72.7 months for tumors with low *TGFB2* expression [11]. In low-grade gliomas (LGGs), elevated mRNA levels of *TGFB2* emerged as a significant negative prognostic marker, particularly when coupled with the activation of interferon-gamma receptor signaling via interferon regulatory factor 5 (*IRF5*) and the increased expression of the immune checkpoint molecule B7-H3 [12].

We have now extended these studies to examine the role of *TGFB2* mRNA in breast cancer prognosis. Published studies have shown that the roles of transforming growth factor isoforms *TGFB1*, *TGFB2*, and *TGFB3* in breast cancer prognosis are context-dependent [13,14,15,16,17,18]. Analysis revealed that both univariate (*p* = 0.01) and multivariate (*p* = 0.013) models showed serum TGFB1 levels that have a positive prognostic impact on overall survival in breast cancer patients [17]. Additionally, a study indicated that patients lacking TGFB1 expression experienced a 4.6 times higher risk of distant recurrence [16]. A comparative analysis of TGFB1 and TGFB3 immunoreactivity demonstrated co-expression of both isoforms in 111 tumors (72.5%), while 16 cases (10%) exhibited no expression. Immunostaining revealed that TGFB3, but not TGFB1, was inversely associated with overall survival (*p* = 0.0204). Furthermore, when considered in conjunction with lymph node involvement, TGFB3 emerged as a significantly more prognostic indicator for overall survival (*p* = 0.0003) [18]. The gene expression of TGFB peptides and receptors were analyzed in both normal and malignant breast tissue, including primary epithelial and stromal cultures. All TGFB isoforms and their receptors were expressed in normal breast samples, which expressed *TGFB1* and *TGFB3* (88% and 89%), while *TGFB2* was less frequent (68%). Tumor stromal cells showed higher *TGFB1* production than normal cells (*p* < 0.0001). In tumor-derived stromal cultures, *TGFB1* and *TGFB2* levels were strongly correlated (r = 0.976) [15]. Investigation of a total of breast cancer biopsy specimens using immunohistochemical analysis assessing expression levels of TGFB receptors *TGFBR1* and *TGFBR2* showed that *TGFBR1* expression was detected in 273 out of 555 samples (49%), while *TGFBR2* expression was observed in 239 out of 474 cases (50%) [19].

We utilized a bioinformatic-driven approach to characterize the impact of *TGFB2* mRNA, in combination with potential prognostic markers, on overall survival in breast cancer patients from The Cancer Genome Atlas (TCGA) database. We implemented a multivariate Cox proportional hazards model to directly compare hazard ratio (HR) calculations for *TGFB2* mRNA, an mRNA product of a marker gene (Gene2) expression, including an interaction term of *TGFB2* by marker gene expression while controlling for age at diagnosis and breast cancer subtypes and comparing patients who received chemotherapy-only (chemo-only) therapies, and we filtered genes that reported HR greater than 1 for patients receiving only chemotherapy in the multivariate model. The interaction term in the model enabled the identification of *TGFB2*/Gene2 mRNA combinations that result in synergistic improvements for breast cancer patients. This resulted in *TGFB2*-dependent positive prognostic markers that could be potentially used as inclusion criteria in biomarker-guided clinical trial designs. We also identified *TGFB2*-independent Gene2, showing a negative correlation between mRNA expression and OS across all mRNA expression levels, and validated it with an independent dataset from the KMplotter database. The increased HR for patients with high levels of expression of these marker genes suggests that these marker genes could be presented as a future development for targeted therapies.

## 2. Results

### 2.1. TGFB2-Dependent Positively Prognostic Gene2

*TGFB2* expression increased 1.95-fold in tumor tissues (Figure 1, *p* < 0.0001), indicating its involvement in tumor pathology. Using multivariate Cox analysis, we found that *TGFB2* mRNA levels interact with Gene2 to affect the HR for overall survival (OS). This analysis was controlled for age at diagnosis, breast cancer subtype, treatment (chemotherapy-only (chemo-only) versus all other treatments), and the interaction between *TGFB2* by Gene2 for 786 evaluable patients. Out of the 15,947 Gene2 screened, 380 genes exhibited significant interaction terms (*p* < 0.05), of which 111 genes exhibited significant upregulation in tumor tissues (Appendix A, *p* < 0.0001, FDR < 0.001).

In our first screen, we sought to identify *TGFB2*-dependent Gene2 effects that exhibited significant HR calculations for *TGFB2* mRNA expression (*p* < 0.05) AND Gene2 expression (*p* < 0.05) AND *TGFB2* by Gene2 interaction terms (*p* < 0.05) (Figure 1; 11 Gene2 upregulated in tumors). Expression of collagen type x alpha 1 chain (*COL10A1*) mRNA exhibited a significant (*p* < 0.0001) 877.9-fold change in tumor compared with normal tissue; this high value for fold change was a result of the very low expression of *COL10A1* in normal tissue (mean ± SEM = −3.92 ± 0.16 log_2_TPM (transcripts per million)). Three genes showed a greater than 3-fold increase in expression in tumors compared with normal tissue: integrin subunit alpha 11 (*ITGA11*), sulfatase 1 (*SULF1*), and hydroxysteroid 17-beta dehydrogenase 6 (*HSD17B6*), all with *p* < 0.0001 (Figure 1B).

Examination of the multivariate Cox proportional hazards model revealed two sets of Gene2 markers from the first screen (Appendix A). Six of these genes showed an increase in HR even when considering the positive prognostic impact of *TGFB2* mRNA and significant interaction terms in the Cox model (Appendix A): *HSD17B6*; *ITGA11*; glypican 4 (*GPC4*); *COL10A1*; ANTXR cell adhesion molecule 1 (*ANTXR1*); and *SULF1* (HR ranged from 1.43 to 1.22). Five genes showed a decrease in the HR of both *TGFB2* and Gene2 using the multivariate Cox proportional hazards model that is multiplied, as suggested by the significant *TGFB2* by Gene2 statistical interaction effects: armadillo repeat containing 7 (*ARMC7*); transmembrane protein 14B (*TMEM14B*); autocrine motility factor receptor (*AMFR*); arylformamidase (*AFMID*); and ganglioside-induced differentiation-associated protein 1 (*GDAP1*), with the HR ranging from 0.02 to 0.62 (Appendix A).

We next focused on Gene2 expression that exhibited positive prognostic impact in combination with *TGFB2* mRNA expression from the list of 380 Gene2s that showed significant interaction parameters from the multivariate Cox proportional hazards models (Appendix A). Seven Gene2s: *GDAP1*, TBL1X/Y related 1 (*TBL1XR1*), ring finger protein, transmembrane 1 (*RNFT1*), 2-hydroxyacyl-CoA lyase 1 (*HACL1*), solute carrier family 27 member 2 (*SLC27A2*), notchless homolog 1 (*NLE1*), and thioredoxin domain-containing 16 (*TXNDC16*) showed significant improvements in OS outcomes when comparing *TGFB2*^high^/Gene2^high^ versus *TGFB2*^high^/Gene2^high^-expressing cohorts of patients using a Kaplan–Meier analysis; this was shown by separation for two arms of the four survival curves (*p* < 0.05), whereby high levels of *TGFB2* and Gene2 mRNA expression resulted in the most improved OS outcomes (Figure 2, Appendix A). Calculation of the multivariate parameters for these seven Gene2 reported *p*-values for the interaction term ranging from 0.004 to 0.036 (Appendix A). Kaplan–Meier analysis of patient cohorts with above median expression of *TGFB2* and *GDAP1* mRNA (*n* = 181; *p* = 0.0084, Figure 2A); *TBL1XR1* mRNA (*n* = 222; *p* = 0.0121, Figure 2B); *RNFT1* mRNA (*n* = 190; *p* = 0.0164, Figure 2C); *HACL1* mRNA (*n* = 184; *p* = 0.0188, Figure 2D); *SLC27A2* mRNA (*n* = 177; *p* = 0.03, Figure 2E); *NLE1* mRNA (*n* = 196; *p* = 0.031, Figure 2F); and *TXNDC16* mRNA (*n* = 185; *p* = 0.036, Figure 2G) showed significantly improved survival outcomes than patients cohorts expressing high mRNA levels of *TGFB2* and low levels of Gene2 (Figure 2, Appendix A). Four of these genes were significantly upregulated in tumor tissues (*p* < 0.0001 for all comparisons) (Figure 2H): *SLC27A2*, *TXNDC16*, *TBL1XR1*, and *GDAP1*.

We investigated the correlation of *TGFB2* with gene signature proteins using the network analyses provided in the protein–protein interaction networks from the STRING database (Figure 3, Figure 4 and Figure 5). We integrated data from two screens analyzing *TGFB2*-dependent Gene2 prognostic impacts (Figure 1 and Figure 2), showing TGFB2 was associated with the network of GPC4, SULF1, ANTXR1, ITGA11, and COL10A1 via linker proteins via second shell interactors: biglycan (BGN), collagen type III alpha 1 chain (COL3A1), collagen type I alpha 2 chain (COL1A2), collagen type V alpha 2 chain (COL5A2), and thrombospondin 2 (THBS2) (Figure 3). The PAM50 gene signature was associated with TGFB2 via MYC proto-oncogene, BHLH transcription factor (MYC), and epidermal growth factor receptor (EGFR) (Figure 4). TGFB2 was associated with the network of the OncotypeDX gene signature via TGFB1 associations with CD68 molecule (CD68), progesterone receptor (PGR), estrogen receptor 1 (ESR1), BCL2 apoptosis regulator (BCL2), and erb-B2 receptor tyrosine kinase 2 (ERBB2) (Figure 5).

### 2.2. TGFB2-Independent Gene2 Correlated with Increase in HR

We next performed multivariate analyses using the Cox proportional hazards model to assess the individual effects of *TGFB2* and Gene2 marker mRNA expression levels, identifying Gene2 that exhibited an increase in HR, independent of *TGFB2* mRNA expression. This analysis was controlled for age at diagnosis, breast cancer subtype, the confounding effect of treatment (chemo-only versus all other treatments), and the interaction between *TGFB2* and Gene2. Out of the 15,947 Gene2 screened, 1286 exhibited increases in HR for Gene 2 (*p* < 0.05), of which 118 genes exhibited increases in HR for Gene2 not impacted by *TGFB2* expression (*p*-value for *TGFB2* > 0.1 AND *p*-value for *TGFB2* by Gene 2 interaction > 0.1 AND chemo-only parameter > 1.5 standard deviation units). We compared the mRNA expression in tumor tissues of 118 Gene2 markers relative to normal tissues, resulting in 41 genes being significantly upregulated in tumor tissues for 786 evaluable patients (Appendix A, Appendix A; using the filter: *p* < 0.0001, fold increase > 1, log_2_ (TPM) expression in tumor tissue > 2).

We cross-referenced the 41 genes identified in the TCGA dataset with those reported in the KMplotter database to narrow down the potential list of prognostic markers using more stringent criteria, comparing only median cut-off values for high- versus low-expressing patient subsets. This analysis identified six significantly upregulated genes in breast cancer tumor tissues cross-referenced in both the TCGA and KMplotter datasets (Figure 6). A significant increase in mRNA expression levels was observed in tumor tissue compared with normal tissue for the following genes: *TAGLN2* (2-fold, *p* < 0.0001), *ENO1* (1.49-fold, *p* < 0.0001), *GLRX2* (1.73-fold, *p* < 0.0001), *PLOD1* (1.72-fold, *p* < 0.0001), *PRDX4* (1.81-fold, *p* < 0.0001), and *TMED9* (2.28-fold, *p* < 0.0001) (Figure 6B).

To predict how Gene2 mRNA expression affects survival, we generated survival curves at any given time by plotting and calculating the shift in the baseline OS curve for 789 breast cancer patients (59 death events) by applying the parameters from the output of the multivariate regression model and then utilizing the fitted parameters (Appendix A) to shift the baseline curves using values fixed for the control variables and varying a single variable of interest, thereby evaluating the Gene2 mRNA-dependent impact on OS. The shift in the baseline survival curves represented the median OS times throughout the entire range of Gene2 expression levels (minimum, lower quartile, median, upper quartile, and maximum), as well as the magnitude of the HR calculation. These simulated survival curves measured the impact of Gene2 alone, accounting for both the range of mRNA expression and the hazard ratio calculations (Figure 7, Table 1). Five genes exhibited HRs greater than 1.3 (30% increase in hazard for one standard deviation unit (Zscore) increase in mRNA expression): *ENO1*, *GLRX2*, *PLOD1*, *PRDX4*, *TMED9*, and *TAGLN2* showed an HR of 1.7, controlling for *TGFB2* mRNA expression, chemotherapy treatment, age at diagnosis, and subtypes of breast cancer (Figure 7, Table 1).

We next validated the TCGA prognostic markers, which was conducted using the independent KMplotter dataset for breast cancer patients with a larger sample size. The analysis focused on six genes (*ENO1/ENO1L1*, *GLRX2*, *PLOD1*, *PRDX4*, *TAGLN2*, and *TMED9*), which were found to significantly influence OS in both the TCGA and KMplotter datasets (Figure 8). At the median cut-off of mRNA expression for Gene2, high levels of expression showed worse OS outcomes for the six gene products identified in the TCGA and KMplotter datasets.

## 3. Discussion

In our first screen, the expression of 11 Gene2 markers exhibited significant prognostic impact for *TGFB2* mRNA expression (*p* < 0.05) AND Gene2 expression (*p* < 0.05) AND *TGFB2* by Gene2 interaction terms (*p* < 0.05). Six of these genes showed an increase in HR even when considering the positive prognostic impact of *TGFB2* mRNA and significant interaction terms in the Cox model: *HSD17B6*; *ITGA11*; *GPC4*; *COL10A1*; *ANTXR1*; and *SULF1*. Notably, expression of *COL10A1* mRNA exhibited a significant increase in tumor tissues, suggesting that expression of this gene is highly tumor-specific. The published literature has reported that in colorectal cancer, *COL10A1* expression is associated with tumor progression and metastasis [20], and high expression of *COL10A1* has been linked to poor overall survival and disease-free survival (DFS) in colorectal cancer patients [21]. In lung cancer, *COL10A1* expression is associated with tumor progression and metastasis [22]. *COL10A1* expression was found to be an independent risk factor for gastric cancer [23]. Interestingly, the circulating expression level of *COL10A1* is significantly increased in gastric adenocarcinoma patients and associated with poor survival in these cancers [23]. Survival analysis of pancreatic cancer patients suggested that high expression of *COL10A1* was associated with a poorer prognosis and that knockdown of *COL10A1* inhibited the proliferation, migration, and invasion of cells in functional assays [24].

*SULF1* mRNA exhibited a significant increase in breast tumor tissue. In our multivariate Cox proportional hazards model, increasing levels of *SULF1* increased the HR in breast cancer patients (22% increase for one standard deviation increase in gene expression), in contrast to the positive prognostic impact of *TGFB2* mRNA levels. Network analysis of SULF1 exhibited a second-order connection to TGFB2 via BGN, COL3A1, COL1A2, COL5A2, and THBS2 proteins.

*SULF1* is a member of the sulfatase family, which regulates the sulfation of heparan sulfate proteoglycans and has been implicated in the prognosis of various types of cancer for both positive and negative prognostic impact. *SULF1* mRNA expression was associated with a poor prognosis in lung adenocarcinoma [25]. A study that employed quantitative real-time polymerase chain reaction (qRT-PCR) to measure *SULF1* mRNA expression from 54 non-small-cell lung cancer (NSCLC) patients showed that patients expressing high levels of *SULF1* mRNA exhibited shorter median OS times. Furthermore, knockdown of *SULF1* suppressed key malignant behaviors in NSCLC cell lines, including NCI-H1299 (CRL-5803) and HCC827 (CRL-2868), by reducing cell proliferation, migration, invasion, and epithelial–mesenchymal transition (EMT) through inhibition of the EGFR/MAPK signaling pathway [26]. In hepatocellular carcinoma (HCC), *SULF1* promotes TGFB-induced gene expression and EMT transition [27]. In vitro studies demonstrated that forced *SULF1* expression in HCC cell lines (Hep3B, PLC/PRF/5) resulted in increased SMAD2/3 phosphorylation following stimulation with TGFB1. Conversely, *SULF1* knockdown in cell lines with high endogenous levels (SNU182, SNU475) reduced TGFB signaling, migration, and invasiveness [27]. Furthermore, both immunohistochemistry (IHC) and Western blotting identified elevated *SULF1* expression and increased phosphorylation of SMAD2/3 (TGFB pathway activation) in tumor and peritumoral tissues from transgenic Sulf1-Tg mice compared with WT [27]. *SULF1* expression facilitated the upregulation of the hallmarks of EMT mesenchymal markers: N-cadherin, vimentin, and αSMA [27]. Notably, these observations were based on the activation of *TGFB1* expression, whereas our prognostic model suggested that the *TGFB2* mRNA isomer drove a positive prognostic impact in breast cancers, indicating the need for a careful and nuanced examination of the TGFB ligand isoforms in tumor progression. In gastric cancer, [28] Fang et al., 2024, reported that *SULF1*, produced by cancer-associated fibroblasts (CAFs), facilitated metastasis and resistance to cisplatin treatment via TGFB1 activation of TGFBR3-mediated signaling. Pan-cancer analysis revealed that *SULF1* mRNA was upregulated more than 2-fold in 16 out of the 32 cancers in the TCGA dataset [29]. Analysis of a single-cell RNA-seq (scRNA-seq) dataset of head and neck squamous cell carcinoma (HNSC) showed the highest positivity of *SULF1* in fibroblasts [29]. In pancreatic cancer, high *SULF1* expression has been associated with later T, N, and TNM stages, higher CA19-9 levels, smaller tumor size, and poorer prognosis [30].

There is a tumor suppressor role for SULF1, being an extracellular sulfatase that removes 6-O-sulfate groups from heparan sulfate (HS) chains, thereby reducing the activity of HS-binding growth factors such as FGF2, VEGF, amphiregulin, HB-EGF, and HGF [31]. By removing these sulfate groups, SULF1 decreases growth factor presence at the cell surface and restricts receptor interaction, ultimately suppressing pro-tumorigenic signaling [31].

In breast cancers under hypoxic conditions, *SULF1* is downregulated by HIF-1α, promoting cancer cell migration and invasion, in aggressive breast cancer cell lines [32]. In this study, a Kaplan–Meier survival analysis showed that tumors with high *SULF1* mRNA expression exhibited longer DFS compared with patients whose tumors had low levels of *SULF1* mRNA expression [32]. Ovarian cancer cell lines and primary tumors lacking *SULF1* mRNA expression showed dense DNA methylation in 12 CpG sites within exon 1A and increased histone H3 methylation around the *SULF1* gene promoter, leading to transcriptional repression [33]. Downregulation of *SULF1* mRNA in ovarian cancer cells, achieved via siRNA, reduces sensitivity to cisplatin-induced cytotoxicity, indicating that loss of *SULF1* mRNA promotes chemoresistance. Furthermore, loss of *SULF1* leads to decreased expression of the pro-apoptotic protein Bim via increased ERK signaling, promoting tumor cell survival and resistance to chemotherapy [34].

Taken together, the impact of *SULF1* in the TME is highly context-dependent on the tumor type, expression in stromal cells, the mechanism of *SULF1* gene activation, and hypoxia. *SULF1* interacts with the TGFB pathway via TGF-beta 1. Our analysis’s high levels of *SULF1* are overexpressed in tumors and have a negative prognostic impact in the context of low levels of a different TGFB isomer, *TGFB2*, suggested by the significant statistical interaction term calculated from the multivariate Cox proportional hazards term. The network analysis showed no direct mechanistic connection of TGFB2 and SULF1, but they are components of a highly interconnected set of associations. Further work will need to elucidate the expression of *TGFB1*, *TGFB2*, and *SULF1* using scRNAseq studies for the distribution of expression in the cellular compartments in tumors.

Expression of *ITGA11* mRNA exhibited a significant (*p* < 0.0001) 3.88-fold increase in tumor tissues. *ITGA11* mRNA expression has been found to play a significant role in cancer prognosis. Studies have shown that high expression of *ITGA11* is associated with poor prognosis in various types of cancer, including gastric cancer [35], NSCLC [36], and breast cancer [37]. In breast cancer, *ITGA11* has been found to play a role in the regulation of cancer-associated fibroblasts, tumor progression, and prognosis [37,38].

Five genes showed positive prognostic impact of both *TGFB2* and Gene2 in that combination is enhanced as suggested by the significant *TGFB2* by Gene2 interaction effects: *ARMC7*; *TMEM14B*; *AMFR*; *AFMID*; and *GDAP1*. Our analysis revealed a novel positive *TGFB2*-dependent effect of these Gene2 markers on breast cancer survival, whereas previous studies on the expression of these *TGFB2*-dependent genes focused solely on their impact on either the risk score or as negative prognostic biomarkers.

*ARMC7* is one of the seven differentially expressed mRNAs included in a prognostic signature for predicting recurrence and DFS in cervical cancer patients [39]. High plasma levels of a five-gene signature, which included *ARMC7*, were significantly associated with inferior overall survival in diffuse large B-cell lymphoma patients, independent of the National Comprehensive Cancer Network International Prognostic Index score [40].

*AMFR* has been shown to have significant prognostic implications in various cancers, primarily indicating a poor survival outcome in NSCLC [41], gastric cancer [42], and a borderline statistically significant correlation with shorter overall survival (*p* = 0.0331) in breast cancers [43]. Interestingly, in colorectal cancer *AMFR* overexpression correlated with improved DFS in colon cancer but was associated with decreased DFS in corresponding nodal metastases. In rectal cancer, *AMFR* overexpression was linked to significantly reduced OS, DFS, and disease-specific survival (*p* < 0.001, *p* = 0.031, *p* = 0.005, respectively) [44]. This supports our observation of the context-dependent impact of *AFMR* expression on clinical outcomes in cancer patients.

In our second screen, we identified *TGFB2*-dependent positive prognostic markers from the survival dataset directly by performing a Kaplan–Meier analysis using median cut-offs for *TGFB2* and the paired Gene2 expression (*GDAP1*, *TBL1XR1*, *RNFT1*, *HACL1*, *SLC27A2*, *NLE1*, *and TXNDC16*). Four of these genes’ mRNA were significantly upregulated in tumor tissues (*p* < 0.0001 for all comparisons) (Appendix A; *SLC27A2* (6.78-fold increase); *TXNDC16* (1.85-fold increase); *TBL1XR1* (1.81-fold increase), and *GDAP1* (1.53-fold increase)).

GDAP1 is primarily involved in mitochondrial function and is localized to the outer mitochondrial membrane [45]. A high level of *GDAP1* expression was correlated with improved survival in pancreatic cancer patients (HR: 0.71 [95% CI: 0.57–0.90]; *p* = 0.004) [46]. This contrasts with the negative prognostic marker in acute myeloid leukemia [47]. *GDAP1* contributed to a gene signature panel correlated with risk prediction in hepatitis B virus-associated HCC [48]. Our studies indicated that the prognostic impact was *TGFB2* mRNA-dependent using both the Cox proportional hazards model and the Kaplan–Meier analysis, suggesting that breast cancer patients with high levels of both *TGFB2* and *GDAP1* expression displayed significantly improved OS outcomes. Therefore, patients expressing high mRNA levels of these two genes displayed improved OS outcomes to standard therapies, which requires confirmation in prospective clinical trials.

SLC27A2 is primarily involved in the cellular uptake and activation of long-chain and very long-chain fatty acids in human tissues [49]. In breast cancers, *SLC27A2* mRNA expression constituted part of a four-gene signature that stratified patients into low and high-risk groups [50]. Increased levels of *SLC27A2* were validated at both the mRNA and protein levels in breast cancer cell lines, and the knockdown of *SLC27A2* inhibited cell proliferation and cell cycle in these cells [50]. The oncogenic role of *SLC27A2* was demonstrated in differentiated thyroid cancer [51] and acute lymphoblastic leukemia [52]. Expression of *SLC27A2* was found to be context-dependent based on its tumor-suppressive role in renal cell carcinoma [53]. In this cancer, *SLC27A2* influences EMT transition by negatively regulating *CDK3*, whose low expression is associated with a poor prognosis [53]. Our studies also suggested that the prognostic effect of *SLC27A2* was context-dependent on *TGFB2* mRNA levels, whereby high levels of *SLC27A2* and *TGFB2* mRNA levels showed improved OS outcomes.

TXNDC16 is a thioredoxin domain-containing protein that belongs to the larger thioredoxin superfamily, which plays a role in supporting cellular redox homeostasis, proper protein folding, and electron transfer. Members of the thioredoxin superfamily of proteins have been reported to be influential in glioma pathogenesis [54]. TBL1XR1 is a multifunctional protein, and its functions are cell type- and context-dependent. It is understood to be involved in transcription and a range of signaling pathways [55]. A recent review identified 12 different cancers, including breast cancer, which implicate *TBL1XR1* in cancer progression [56]. In breast cancer, *TBL1XR1* was correlated with clinical stage, tumor classification, node classification, metastasis classification, and histological grade, and higher *TBL1XR1* expression was an independent prognostic indicator for overall survival. The mechanism may be via cyclin D1-transactivation and activation of the β-catenin signaling pathway [55].

Three genes (*NLE1*, *HACL*, and *RNFT1*) did not exhibit significant upregulation in tumor tissues. In contrast to the positive prognostic impact of these genes in combination with *TGFB2*, *NLE1* is elevated and positively correlated with tumor stage and metastasis in melanoma [57]. Up-regulation of the *NLE1* mRNA in colon cancer cells promotes cell migration and invasion of colon cancer cells and inhibits the apoptosis ability of colon cancer cells [58]. *NLE1* has also been implicated in the progression of non-small-cell lung cancer. *NLE1* expression is significantly higher in tumor tissues than in normal tissues, is correlated with the pathological stage, and can be considered a tumor promoter [59]. Finally, in HCC, high *NLE1* expression has been shown to promote cell proliferation and to be strongly associated with poor prognosis [60].

Investigation of the *TGFB2*-dependent gene signatures for their protein–protein associations from the two screens in our study demonstrated that TGFB2 was associated with a network consisting of GPC4, SULF1, ANTXR1, ITGA11, and COL10A1 via the linker proteins BGN, COL3A1, COL1A2, COL5A2, and THBS2. Studies have demonstrated that TGFB induces BGN expression in pancreatic cells through the activation of MKK6-p38 MAPK signaling downstream of Smad signaling, offering an insight into the regulation of BGN observed in fibrosis and desmoplasia associated with inflammatory responses [61]. Furthermore, the TGF-beta/ALK5 effect on p38 activation and *BGN* expression was also impacted by overexpression of GADD45beta alone in PANC-1 and osteosarcoma MG-63 cells [62]. Regarding research directed towards identifying direct Smad targets in dermal fibroblasts following TGFB stimulation, requiring demonstration of a rapid increase in mRNA levels, TGFB-induced promoter activity blocked by dominant-negative Smad3 and Smad7 vectors, and no promoter transactivation in Smad3(-/-) fibroblasts found *COL3A1* as a TGFB/Smad3 target [63]. TGFB2 treatment elevates *COL1A2* expression in human epithelial SRA01/04 cells, whereby *COL1A2* knockdown inhibited TGFB2-induced SRA01/04 cell proliferation, migration, invasion, and EMT [64]. *COL5A2* expression was shown to be elevated in human osteosarcoma cells, and the downregulation of *COL5A2* affected the TGFB and Wnt/β-catenin signaling pathways [65]. *THBS2* expression in pancreatic cancer is mainly present in the stroma and is linked to tumor progression and poor prognosis. RNA in situ hybridization revealed that cancer-associated fibroblasts (CAFs), not tumor cells, express *THBS2*, with levels increasing as the disease progresses in mouse models [65]. TGFB1 from cancer cells activated CAFs to produce *THBS2* through the p-Smad2/3 pathway, suggesting an important role for TGFB association with *THBS2* in cancer progression [66]. It is noted that the TGFB2 association represents a strong correlation; future controlled experiments in reduced systems will be required to infer mechanistic insight of this TGFB2 association in breast cancer tissues.

Since our multivariate prognostic model identified *TGFB2*-dependent prognostic markers across the TCGA cohorts including unclassified, basal, normal, luminal A, luminal B, and HER2+, we performed a STRING network analysis on the PAM50 gene signature that classifies cancers according to luminal A, luminal B, HER2+, basal-like, and normal-like, and it has been further developed by combining risk scores to improve future recurrence risk [67,68]. Our analysis revealed that TGFB2 was linked to the network of the PAM50 gene signature through MYC and EGFR. In cancer cell lines, studies have demonstrated that genetic or pharmacological inhibition of MYC in MCF10A basal breast cells results in increased sensitivity to TGFB-stimulated invasion and metastasis [69], and signaling via Smad3 and ERK/Sp1 mediate TGFB-induced EGFR upregulation [70]. In particular, from our studies, the mRNA expression of *GPC4*, *SULF1*, *ANTXR1*, *ITGA11*, and *COL10A1* exhibited significant statistical interactions with *TGFB2* mRNA. *TGFB2* was a positive prognostic factor, and the five genes were negative prognostic factors from the HR calculations of multivariate models (Appendix A). Our results suggest that combinations of high levels of *TGFB2* mRNA and low levels of these five genes resulted in improving patient OS outcomes. This correlation suggests that *TGFB2* mRNA expression levels in breast cancer tumors may be used in conjunction with the PAM50 gene signature to assess the prognostic impact on patients following validation in prospective clinical trials.

OncotypeDx is a genomic test used for early-stage, hormone receptor-positive, HER2-negative breast cancer to help guide treatment decisions, specifically regarding chemotherapy. It analyzes the expression of 21 genes in a tumor sample to provide a recurrence score from 0 to 100. This score indicates the risk of cancer returning and the potential benefit from chemotherapy [71]. The STRING network analysis showed associations of this gene signature with TGFB2 via TGFB1 associations with CD68, PGR, ESR1, BCL2, and ERBB2. TGFB1, ERBB2, and CD68 form a signaling network that impacts cancer progression, invasion, and resistance to therapy in HER2-overexpressing breast cancer tumors [72,73]. There was no direct association of *TGFB2* mRNA and the OncotypeDX gene signature.

Expression of six genes (*ENO1*, *GLRX2*, *PLOD1*, *PRDX4*, *TAGLN2*, and *TMED9*) was identified to be significantly upregulated in tumor tissues, exhibiting an increase in HR on breast cancer patients analyzed in the TCGA dataset using the multivariate Cox proportional hazards model and validated in an independent KMplotter dataset.

In breast cancer, *ENO1*’s role can be context-dependent: in early-stage breast cancer, higher *ENO1* expression correlates with better survival and increased anti-tumor immune infiltration, while in advanced disease, it is associated with progression and immune evasion [74]. *ENO1* is frequently overexpressed and is consistently linked to poor prognosis, advanced stage, and aggressive tumor features across multiple malignancies, including breast, lung, bladder, prostate, gastric, colorectal, pancreatic, and thyroid cancers [75,76,77,78]. Mechanistically, *ENO1* drives tumorigenesis by promoting glycolysis (the Warburg effect); activating oncogenic signaling pathways such as PI3K/Akt, β-catenin, TGFB/Smad, and Wnt; and facilitating epithelial–mesenchymal transition, angiogenesis, and immune evasion [79,80,81,82,83,84]. *ENO1* also regulates apoptosis, cell cycle progression, and interacts with c-Myc [85]. ENO1’s cell surface localization and cancer-specific overexpression make it a promising and accessible biomarker for diagnosis and prognosis, as well as a therapeutic target. Preclinical studies demonstrate that targeting ENO1 using antibodies, small molecule inhibitors, DNA vaccines, or gene knockdown can suppress tumor growth, invasion, metastasis, and chemoresistance and may enhance the efficacy of chemotherapy [78,79,84,86,87]. Additionally, *ENO1* influences the TME by modulating immune cell infiltration and promoting CD8+ T cell exhaustion in some cancers [76,88]. *ENO1* is a central driver of cancer metabolism, progression, and immune modulation, making it a promising candidate for integrated diagnostic, prognostic, and therapeutic strategies in oncology [74,77,78]. Availability of a small, cell-permeable molecule inhibitor of ENO1 such as ENOblock/AP-III-a4 [89] suggests that breast cancer patients with high levels of *ENO1* mRNA can be potentially targeted for treatment. ENO1 inhibition enhances anti-tumor immunity and sensitizes tumors to immune checkpoint blockade therapies, such as anti-PD-L1, in bladder cancer models. CRISPR-Cas9-mediated knockout of *ENO1* and ENOblock reduced tumor burden and promoted infiltration of cytotoxic CD8+ T cells [90]. A citrullinated peptide vaccine targeting ENO1 (citENO1 vaccine) in triple-negative breast cancer (TNBC) mouse models showed synergistic effects with PD-1 blockade, significantly inhibiting tumor growth and improving survival compared with controls [91]. Our results suggest that *ENO1* is targetable for treated patients with high levels of *ENO1* mRNA. The prognostic impact at the mRNA level will need to be confirmed by measuring protein levels in the tumors.

GLRX2 maintains mitochondrial redox balance by modulating glutathionylation and interacting with thioredoxin/glutathione systems, protecting against oxidative stress-induced apoptosis [92]. It is implicated in a variety of other cancers. High *GLRX2* expression correlates with better prognosis and survival in early-stage colon adenocarcinoma (87% in stage I vs. 1% in stage III), acting as an independent prognostic marker [93]. *GLRX2* is upregulated in bladder cancer and contributes to diagnostic models, suggesting a pro-tumorigenic role [94]. In gliomas, high *GLRX* expression is associated with aggressive subtypes (wild-type *IDH*, methylated *MGMT*) and is involved in immune modulation [95]. Serum GLRX2 levels in colon cancer patients correlate with survival [93]. *GLRX2* influences tumor-associated immune processes, particularly via M0 macrophages and immune checkpoint pathways in gliomas, indicating a role in shaping the TME [95]. Currently, no specific inhibitors to GLXR2 have been described in the literature. Our results suggest that breast cancer patients with low levels of *GLXR2* expression exhibited improved OS outcomes with standard therapies. Prospective clinical trials examining *GLXR2* mRNA levels will be required to validate our observations.

PLOD1 hydroxylates lysine residues in collagen, enabling cross-linking that strengthens connective tissues (e.g., skin, bones). *PLOD1* is upregulated in osteosarcoma, gastric, breast, glioma, bladder, pancreatic, and hepatocellular cancers (HCCs), correlating with advanced tumor stages, metastasis, and reduced survival [96,97,98,99]. High *PLOD1* expression predicts poor overall and DFS in multiple cancers, including breast cancer [100,101,102]), and *PLOD1* inhibition (via siRNA or small molecules) attenuates proliferation, migration, and invasion in bladder cancer and glioma models [97,103]. *PLOD1* drives proliferation, migration, invasion, and therapy resistance in cancers by modulating pathways like β-catenin, NF-κB, and Akt/mTOR [96,97,98]. In HCC, PLOD1 stabilizes SEPT2 via hydroxylation, promoting F-actin networks that enhance confined cell migration [99]. *PLOD1* expression correlates with the infiltration of immunosuppressive cells (e.g., macrophages, fibroblasts) and reduced CD8+ T cell activity. It is also associated with immune checkpoints such as PD-L1 [100,104]. PLOD1 protein contains a prolyl 4-hydroxylase alpha subunit homologs (P4Hc) domain that can be targeted for inhibition by compounds akin to FG-4592 (roxadustat), GSK1278863 (daprodustat), molidustat, IOX4, and vadadustat [105]. Our study suggests that patients expressing high levels of PLOD1 can be targeted for therapies against prolyl 4-hydroxylase alpha subunits pending confirmation that an increase in *PLOD1* mRNA results in increases in PLOD1 protein levels in breast cancer patients.

PRDX4 is a key endoplasmic reticulum (ER)-resident antioxidant enzyme involved in hydrogen peroxide metabolism, redox homeostasis, and oxidative protein folding, thereby protecting cells from oxidative and ER stress and supporting normal physiological processes such as protein folding and ovarian function [106,107,108,109]. In cancers, *PRDX4* is frequently overexpressed in multiple tumor types, including breast, prostate, ovarian, colorectal, lung, and gastric cancers, and is associated with tumorigenesis, therapeutic resistance, metastasis, and recurrence [106,107,108,109,110]. Mechanistically, *PRDX4* promotes cancer cell proliferation, migration, invasion, and radio resistance, often through pathways such as Akt/GSK3 and by modulating the TME [107,109,110]. Knockdown or depletion of *PRDX4* in cancer cells leads to reduced proliferation, increased apoptosis, impaired migration and invasion, and heightened sensitivity to radiation and chemotherapy [107,109]. *PRDX4* also influences the immune landscape of tumors. Its overexpression is linked to reduced lymphocyte infiltration and increased immunosuppression, partly through interactions with proteins like TXNDC5 and modulation of immune checkpoints [108]. In vivo studies in a silicosis mouse model demonstrated that pharmacological inhibition of PRDX4 using conoidin A (Con A) significantly improved lung function, reduced fibrosis and inflammatory infiltration, and decreased collagen deposition and lung damage [111]. Our correlative results suggest that breast cancer patients with high levels of *PRDX4* may be targeted using compounds such as Con A, pending confirmation that increased protein levels are also prognostic in breast cancer patients.

TAGLN2 is essential for immune cell function, particularly in T cells and macrophages, where it stabilizes the actin cytoskeleton at the immunological synapse, enhances T cell activation, and promotes phagocytosis [112,113]. *TAGLN2*’s role in breast cancer metastasis was examined through transwell migration, luciferase, and flow cytometry assays, as well as a mouse xenograft model, suggesting *TAGLN2* acts as a tumor suppressor, and its loss may enhance breast cancer metastasis via activation of the ROS/NF-κB pathway [114]. However, our results suggest that upregulation of *TAGLN2* mRNA was correlated with worse prognosis in breast cancer patients and concurred with other studies, including gastric, colorectal, papillary thyroid, and endometrial cancers, whereby *TAGLN2* is overexpressed and correlated with poor prognosis, advanced tumor stages, metastasis, and drug resistance [115,116,117,118]. It promotes proliferation, migration, invasion, angiogenesis, and EMT [117,118,119]. In breast cancer patients, the impact of *TAGLN2* may operate via activation of the NRP1/VEGFR2 and MAPK pathways, as in gastric cancer, promoting angiogenesis and cell survival [119]; the Rap1/PI3K/AKT pathway, as reported in papillary thyroid cancer, to enhance invasion [117]; or, akin to colorectal cancer progression, by activating STAT3 and regulating ANXA2 [118]. A more careful examination of the signaling mechanisms from breast cancer patients will need to be elucidated to clarify the roles of tumor-suppressing and tumor-promoting signaling in these patients. Knocking down *TAGLN2* may present as a target in breast cancer patients, based on reports that *TAGLN2* knockdown increases DNA damage and sensitizes gastric cancer cells to chemotherapy and radiation [116]. Depleting *TAGLN2* inhibits migration and invasion in endometrial cancer, and recombinant *TAGLN2*-based therapies reduce tumor growth in vivo, suggesting therapeutic potential [115]. Small molecule inhibitors that block transgelin-2-dependent actin polymerization have been found to significantly reduce growth and invasion of GBM 08-387 and 3359 PDX cell lines in vitro [120]. Although no clinical trials are testing these compounds, inhibition of *TAGLN2* may be pursued as a therapeutic option for breast cancer patients expressing high levels of this gene following confirmation of the prognostic impact of the protein in breast cancer patients.

*TMED9* is upregulated in breast cancer, HCC, ovarian cancer, and gliomas, correlating with advanced tumor stages, metastasis, and poor survival [121,122,123,124,125]. *TMED9* drives colon cancer metastasis via the CNIH4/TGFα/GLI pathway, inducing EMT and enhancing migration/invasion [125,126]. *TMED9* promotes oncogenic signaling via TGFα/GLI, EMT, and STAT3 pathways [125,126]. In gliomas, *TMED9* correlates with immune cell infiltration (e.g., B cells, T cells) and immune checkpoints like CD274 (PD-L1), suggesting a role in immune evasion [127]. BRD4780 is a small molecule that targets TMED9 [128]. Our results suggest that breast cancer patients with high levels of *TMED* mRNA are potentially susceptible to inhibiting TMED pending confirmation of protein levels.

Applying bioinformatics, our hypotheses-generating study identified biomarkers correlated with OS in breast cancer patients, underscoring the importance of further laboratory validation of the marker mRNA and protein expression of these markers. Our observations suggest that biomarkers predictive at elevated *TGFB2* mRNA levels may be influenced by their biological and biochemical context, warranting additional studies to clarify these relationships. Protein–protein interaction network maps revealed that TGFB2 correlates with EGFR and MYC from the PAM50 gene signature, indicating that protein-level validation of these genes is needed to determine if *TGFB2*-related markers can complement the PAM50 signature in future prospective clinical trials. Further validation will require cell-level expression in the TME to determine the localization of *TGFB2* and the biomarkers, thereby elucidating the establishment of *TGFB2* dependency correlated with OS outcomes. Thus, a validation pipeline combining qRT-PCR, IHC, and scRNA-seq studies will enable comprehensive confirmation of biomarkers by integrating bulk quantitative mRNA measurement, spatial protein localization, and single-cell resolution of tumor heterogeneity. Initial qRT-PCR on bulk tumor RNA quantifies marker mRNA levels and validates differential expression, followed by IHC, to analyze protein expression patterns and distribution in formalin-fixed tissues. Concurrently, scRNA-seq of fresh tumor samples will reveal cell type-specific expression, heterogeneity, and involvement in TME dynamics. Integrating these complementary approaches ensures robust validation by confirming molecular signatures predicted by bioinformatics, correlating mRNA and protein data, and elucidating the roles of biomarkers in distinct cellular subpopulations, thereby accelerating clinical translation and personalized therapy development in breast cancers.

## 4. Materials and Methods

### 4.1. AI-Powered Chatbot Identified PubMed Abstracts for Manuscript Preparation

PubMed searches using the keywords: “(ARMC7 OR TMEM14B OR AMFR OR AFMID OR GDAP1) AND cancer” (171 abstracts); “(GDAP1 OR SULF1 OR ITGA11 OR HSD17B6 OR COL10A1) AND cancer” (410 abstracts); “TGF-beta AND Breast” (3667 abstracts); and “Gene AND TCGA AND prognosis AND breast” (2273 abstracts) (https://pubmed.ncbi.nlm.nih.gov/, accessed on 18 December 2024) were downloaded as text documents for processing using the Oncotelic Chatbot technologies. We also utilized Perplexity AI chatbot to identify articles (https://www.perplexity.ai/, accessed on 27 October 2025) not deposited in PubMed. Detailed methods are described in the Appendix A. Chatbot technology enabled interactive keyword refinement for PubMed keyword searches by conducting question-and-answer sessions to refine the references; however, it was not utilized for systematic review, result validation, hypothesis formulation, or manuscript writing. All biomarker discoveries in the manuscript resulted from bioinformatic analyses, and identification of these genes were used to interrogate PubMed and Perplexity AI tools to contextualize the literature for these genes.

### 4.2. Differential Expression of mRNA Comparing Breast Cancer Tumors Versus Normal Samples

We analyzed gene expression by downloading log_2_-transformed TPM RNA-seq summary files (see https://toil-xena-hub.s3.us-east-1.amazonaws.com/download/TcgaTargetGtex_rsem_gene_tpm.gz (accessed on 10 November 2025) for full metadata) from the UCSC Xena web platform (https://xenabrowser.net/datapages/, accessed on 25 July 2023) [129]. This enabled a comparative study of 179 normal breast tissue samples (“GTEX Breast”) and 786 breast cancer patient samples with associated clinical data (“TCGA Breast Invasive Carcinoma”). The data are sourced from the UCSC Toil RNA-seq recompute compendium, which provides a harmonized dataset with realigned and recalculated gene and transcript expression levels for all TCGA, TARGET, and GTEx samples [130], facilitating direct comparisons of gene expression between TCGA tumor samples and matched GTEx normal tissues.

The gene expression measure in the Xena database, especially for TCGA cohorts, is normalized, log_2_-transformed RSEM values represented as log_2_(x + 1), where x is the normalized expression value estimated by the RNA-Seq Expectation-Maximization (RSEM) method. This metric is designed to facilitate cross-sample comparisons by stabilizing variance and normalizing variations in sequencing depth. We applied ANOVA methods to determine differential gene expression values such as RSEM normalization and transformations for the following features: 1. variance stabilization to make the data more normally distributed, satisfying key ANOVA assumptions; 2. variance modeling by quantifying uncertainty and variance in expression estimates by incorporating read mapping ambiguity and fragment length distributions during expectation maximization; and 3. handling heteroscedasticity by modelling the variance mean dependency explicitly. Log-transforming RSEM values reduces heteroscedasticity (unequal variance), thereby improving the validity of ANOVA or related variance component tests on expression data. There are related methods (DESeq2/edgeR) that leverage count reads directly for gene-level comparisons. The Xena platform provides batch-corrected RSEM-normalized and -transformed expression values from two different datasets, including GTEx normal samples, TCGA tumor samples, and adjacent normal samples, which enables robust comparisons between normal and tumor samples.

We applied a two-way ANOVA model to identify differentially expressed genes to compare normal versus tumor tissue samples. The log_2_-transformed TPM values for gene and tissue were included as fixed factors, along with one interaction term to investigate gene-level effects for normal and tumor tissues (gene × tissue). For each gene, we conducted a comparison between normal and tumor samples blocked by the gene factor and then determined significance by adjusting the *p*-value using the false discovery rate (FDR) algorithm provided for in the R-package (FDR corrected for all pairs) calculations performed in R using multcomp_1.4-17 and emmeans_1.7.0 packages ran in R version 4.1.2 (1 November 2021) with RStudio front end (RStudio 2021.09.0+351 “Ghost Orchid” Release). Bar chart graphics were constructed using the ggplot2_3.3.5 R package.

We applied a two-way hierarchical clustering approach to arrange gene expression profiles, grouping together both samples and genes with comparable mRNA expression patterns. This was achieved by employing the average linkage method and the default Euclidean distance metric, as implemented via the heatmap.2 function from the R package gplots_3.1.1. The resulting cluster visualization illustrated the mean expression in tumor samples, normalized to the average expression observed in normal breast cancer tissue, and presented as log2-transformed fold-change values. Dendrograms were generated for both the rows (genes) and the columns (samples), thereby displaying the organization and relationship of co-expressed genes across the patient cohort.

### 4.3. OS Outcomes for Breast Cancer

We analyzed clinical metadata (https://www.cbioportal.org/study/summary?id=brca_tcga_pan_can_atlas_2018; accessed on 1 February 2024) and RNA sequencing data (“data_mrna_seq_v2_rsem.txt”: Batch normalized from Illumina HiSeq_RNASeqV2 using the RSEM algorithm (Transcripts per million (TPM)) from 789 patients diagnosed with breast cancer to compare OS correlates with mRNA expression values.

Multivariate analyses utilized the Cox proportional hazards model to assess the individual effects of *TGFB2* and Gene2 mRNA expression levels (screened 15,947 genes) on overall survival (OS) to calculate the HR. This analysis was controlled for age at breast cancer subtype, diagnosis, treatment (chemotherapy-only versus all other treatments), and the interaction between *TGFB2* and Gene2 mRNA. The inclusion of a statistical interaction term was an important consideration in our analysis design. Gene expression of *TGFB2* mRNA and Gene2 mRNA were the main effects of the model. The interaction coefficients from the interaction term in the model represents the additional effect of the interaction term at a specific expression level of *TGFB2* mRNA on Gene2 expression levels; the interaction term determines how much more the effect of *TGFB2* mRNA levels impacts the effect of Gene2 mRNA expression. The HR associated with an interaction term is a multiplicative effect on the baseline hazard, meaning it represents a multiplicative change in the HR when both interacting conditions are present, compared with having only the main effects at defined mRNA levels for *TGFB2* and Gene2. Briefly, the model included (i) *TGFB2* mRNA levels as a linear covariate expressed as Zscores (*n* = 789); (ii) the mRNA expression level for Gene2 mRNA levels as a linear covariate expressed as Zscores (*n* = 789); (iii) *TGFB2* by the Gene2 interaction term to determine the dependency of Gene2 prognostic impact on *TGFB2* mRNA levels (*n* = 789); (iv) treatment comparing chemo-only versus other treatment regimens (*n* = 101 for chemo-only; *n* = 688 for other); (v) breast cancer subtype (unclassified as reference (*n* = 78), basal, normal (*n* = 24), luminal A (*n* = 371), luminal B (*n* = 143), and HER2+ (*n* = 50)); and (vi) age at diagnosis expressed as a linear covariate (*n* = 789), implemented in R (survival_3.2-13 ran in R version 4.1.2. Forest plots were utilized to visualize the hazard ratios for Cox proportional hazards models for OS outcomes (survminer_0.4.9 ran in R version 4.1.2 (1 November 2021). The breast cancer subtype factor in the model controlled for differential survival of the patients, thereby identifying gene expression markers with prognostic impact across all the subtypes reported in the database. The life table HRs were estimated using the exponentiated regression coefficient for Cox proportional hazards analyses implemented in R (survival_3.2-13 ran in R version 4.1.2).

To identify potential prognostic Gene2 mRNA markers, we considered only expression of Gene2 that exhibited HR values greater than 1 for chemo-only treated patients. We applied the multivariate Cox proportional hazards model to mRNA expression values across 15,947 genes and considered that Gene2 prognostic impact was dependent on *TGFB2* mRNA levels with the statistical *TGFB2* by the Gene2 interaction parameter *p*-value being less than 0.05. To better understand different combinations of *TGFB2* mRNA levels and Gene2 mRNA expression, we simulated the proportion of patients surviving (number at risk at follow-up time points relative to number at risk at diagnosis) at any given time by plotting and calculating the shift in the baseline OS curve for 789 breast cancer patients (59 death events) using parameters from the output of the multivariate regression model.

### 4.4. OS Analysis Using Kaplan–Meier Comparisons for Genes Identified in the TCGA Dataset

The KMplotter web tool (https://kmplot.com/analysis/index.php?p=service&cancer=breast, accessed on 15 February 2025) was utilized to access ovarian cancer data from the Affymetrix dataset (N = 1879 patients) for the selection of genes identified from the TCGA dataset (by cross-referencing the KMplotter genes (HR > 1 and *p* < 0.01)). Patients were grouped based on the median cut-off for gene expression levels to compare the impact of high versus low expression patient sub-groups (Follow-up threshold = 120 months). The web tool reported the hazard ratio (HR) and log-rank *p*-values [131,132].

Gene expression values were correlated with OS outcomes investigating the impact of mRNA expression of *TGFB2* (median cut-off for high and low levels of expression), further stratified into four groups based on gene expression levels of Gene2 mRNA expression levels (median cut-off for high and low levels) in these patients. We defined *TGFB2* dependency by visualizing the significant interaction effect from the multivariate Cox proportional hazards models using Kaplan–Meier analysis with 50th percentile cut-offs for *TGFB2* and Gene2. This analysis was implemented to show that when comparing high versus low levels of Gene2 expression, the impact on OS was different when comparing high and low levels of *TGFB2* mRNA expression or when comparing high versus low levels of *TGFB2* mRNA; the OS curves were differentially impacted at low and high levels of Gene2. We identified *TGFB2*/Gene2 combinations that exhibited significant curve separation between the two arms of the four survival curves (*p* < 0.05), whereby high levels of *TGFB2* and maker gene mRNA expression resulted in the most improved OS outcomes.

We employed two screens to identify *TGFB2*-dependent markers: 1. in our first screen, the expression of genes that exhibited significant prognostic impact for *TGFB2* mRNA expression (*p* < 0.05) AND Gene2 expression (*p* < 0.05) AND *TGFB2* by Gene2 interaction terms (*p* < 0.05); and 2. the second screen considered Gene2, which showed significant improvement in OS outcomes when comparing *TGFB2*^high^/Gene2^high^ versus *TGFB2*^high^/Gene2^high^-expressing cohorts of patients using Kaplan–Meier analysis.

We identified negative prognostic markers for the whole cohort of breast cancer patients using a Kaplan–Meier analysis of the marker Gene2 that exhibited worse OS times at high levels of Gene2 expression in both TCGA and KMplotter datasets.

### 4.5. Identifying Prognostically Relevant Signaling Proteins Networked for TGFB2-Dependent Gene2, PAM50, and OncotypeDx Signature Genes Using the STRING Interaction Algorithm

Using the STRING interaction algorithm for putative protein–protein interaction (PPI), networks derived from mRNA levels that exhibited significant prognostic impact on the OS were constructed using the STRING version 12 algorithm (https://string-db.org/cgi/input?sessionId=bR8QWNAdLhxe&input_page_show_search=on, accessed 29 October 2025) to identify candidate hub proteins connecting and/or Gene2 expression that potentially networked interactions between proteins coded by the mRNA [133]. In these diagrams, nodes represent protein identifiers while edges illustrate associations between proteins. Network visualization was achieved by seeding inputs from a list of genes that exhibited significant upregulation in tumor tissue and demonstrated prognostic relevance for *TGFB2*-dependent Gene2: the gene list from PAM50 [67] and OncotypeDx [71] gene signatures. The edges depicted the confidence level for each association, determined through multiple sources of experimental evidence, including text mining, laboratory experiments, databases, co-expression, neighborhood analysis, gene fusion, and co-occurrence. Interaction edge scores exceeding 0.4 were used to define protein associations, with the thickness of connecting lines denoting score thresholds of 0.4, 0.7, and 0.9. We investigated the network association of the TGFB2 protein with the *TGFB2*-dependent gene signatures, specifically the PAM50 and OncotypeDx signatures. We identified linking proteins using the capability to identify second-shell proteins with “no more than 20 interactors,” as provided in the “settings” tab of the STRING web portal.

## 5. Conclusions

For *TGFB2*-dependent biomarkers, elevated *TGFB2* mRNA expression is a prognostic biomarker associated with breast cancer patients that is correlated with improved OS outcomes at high expression levels of *GDAP1*, *TBL1XR1*, *RNFT1*, *HACL1*, *SLC27A2*, *NLE1*, and *TXNDC16*. This suggests that patients with high levels of *TGFB2* and Gene2 were correlated with improved OS and used as markers for future prospective clinical trials. A multivariate analysis revealed that when controlling *TGFB2* expression, we identified six genes (*ENO1*, *GLRX2*, *PLOD1*, *PRDX4*, *TAGLN2*, and *TMED9*) that exhibited a negative correlation between mRNA expression and OS across all expression levels in the breast cancer patient, validated with an independent dataset from the KMplotter database. Five of these negative prognostic markers are druggable (*ENO1*, *PLOD1*, *PRDX4*, *TAGLN2*, and *TMED9*). The increased HR for patients with high levels of expression of these Gene2s suggests that these five genes could be presented for targeted therapies following confirmation of protein levels and drug action in breast cancer patients. Examination of the protein–protein interaction networks suggested that the correlation of *TGFB2*-related markers could potentially complement the PAM50 signature in the assessment of OS prognosis in breast cancer patients following validation of the TGFB2/EGFR/MYC protein associations in tumors.

## Figures and Tables

**Figure 1 ijms-26-11580-f001:**
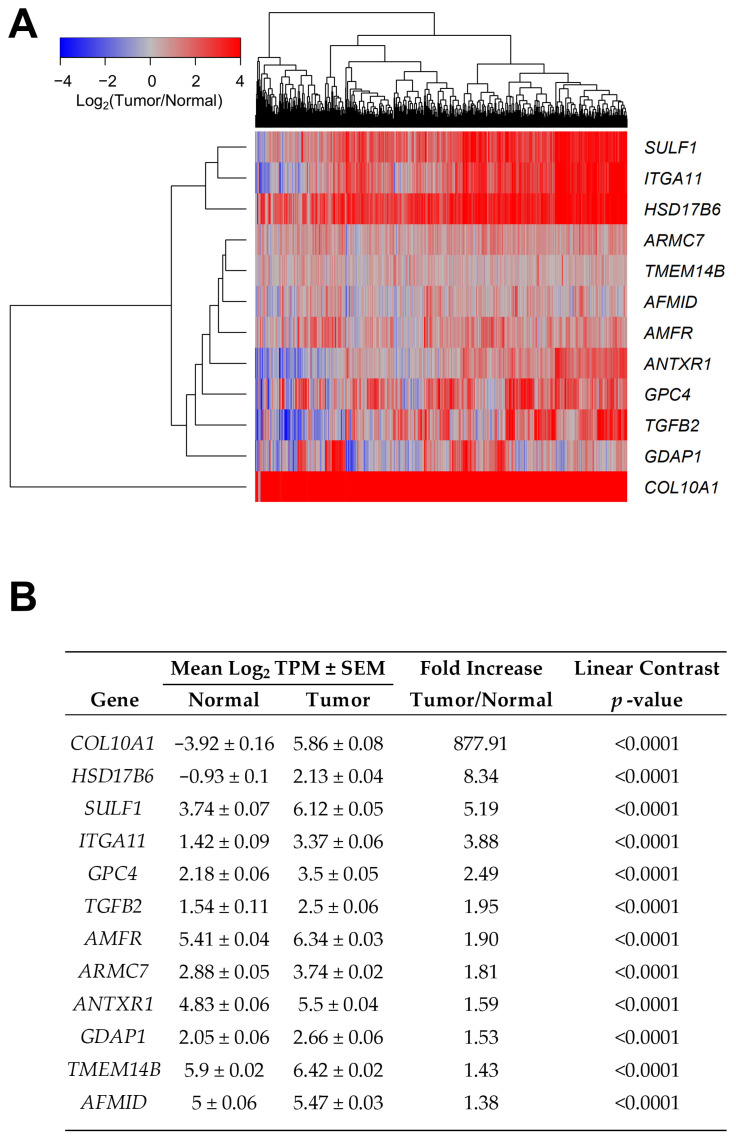
Upregulated genes in tumor tissues with significant prognostic impacts of *TGFB2* mRNA, Gene2, and significant statistical interaction for *TGFB2* and Gene2 expression. Multivariate analyses utilized the Cox proportional hazards model to assess the individual effects of *TGFB2* and Gene2 mRNA expression levels. A total of 111 genes had a significant effect of the *TGFB2* by Gene2 interaction term and exhibited significant upregulation in tumor tissues (*p* < 0.0001, FDR < 0.001). (**A**) The cluster figure shows tumor tissue expression mean centered to normal tissues for 11 out of the 111 genes examined (log_2_ (tumor/normal) TPM), color-coded from blue to red, depicting a decrease to an increase in expression in tumor tissues, respectively; they were further filtered using significant hazard ratio calculations for *TGFB2* mRNA expression (*p* < 0.05) AND Gene2 expression (*p* < 0.05) AND *TGFB2* by Gene2 interaction terms (*p* < 0.05). (**B**) The table depicts a comparison of mRNA expression values, comparing normal (*n* = 179) versus tumor samples (*n* = 786 evaluable samples), ordered by fold increase.

**Figure 2 ijms-26-11580-f002:**
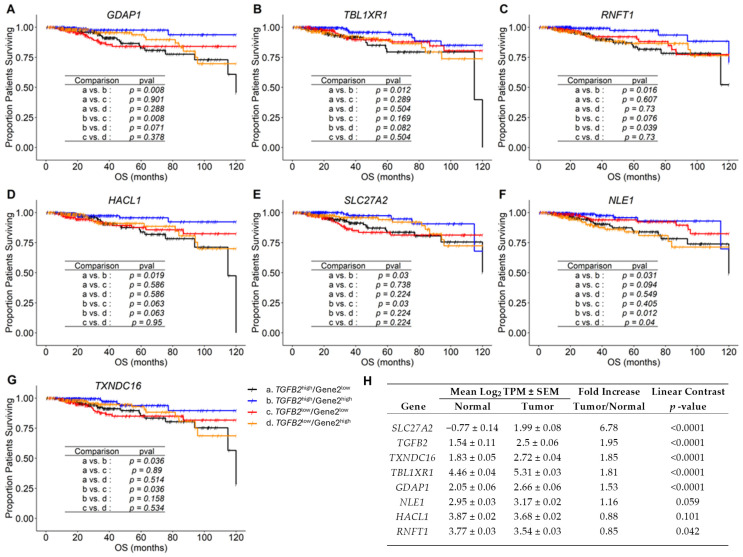
Prognostic markers for breast cancer patients with improved prognosis for high *TGFB2* and Gene2 mRNA expression, assessed using Kaplan–Meier plots. Breast cancer patients were correlated with OS outcomes investigating the impact of mRNA expression of *TGFB2* (median cut-off for high and low levels of expression), further stratified into four groups based on gene expression levels of *GDAP1* (**A**); *TBL1XR1* (**B**); *RNFT1* (**C**); *HACL1* (**D**); *SLC27A2* (**E**); *NLE1* (**F**); and *TXNDC16* (**G**) mRNA expression levels (median cut-off for high and low levels) in these patients. The Kaplan–Meier plots show four stratified curves for each of the Gene2s. Six pairwise comparisons were performed between the four groups of patients (*p*-value adjusted using the Benjamini-Hochberg (BH) correction shown in the table insets). We identified *TGFB2*/Gene2 combinations (a. *TGFB2*^high^/Gene2^low^; b. *TGFB2*^high^/Gene2^high^; c. *TGFB2*^low^/Gene2^low^; d. *TGFB2*^low^/Gene2^high^) that exhibited significant curve separation between the two arms of the four survival curves (*p* < 0.05), whereby high levels of *TGFB2* and maker gene mRNA expression resulted in the most improved survival outcomes. (**H**) The table depicts comparisons of mRNA expression values comparing normal (*n* = 179) versus tumor samples (*n* = 786 evaluable samples), ordered by fold increase.

**Figure 3 ijms-26-11580-f003:**
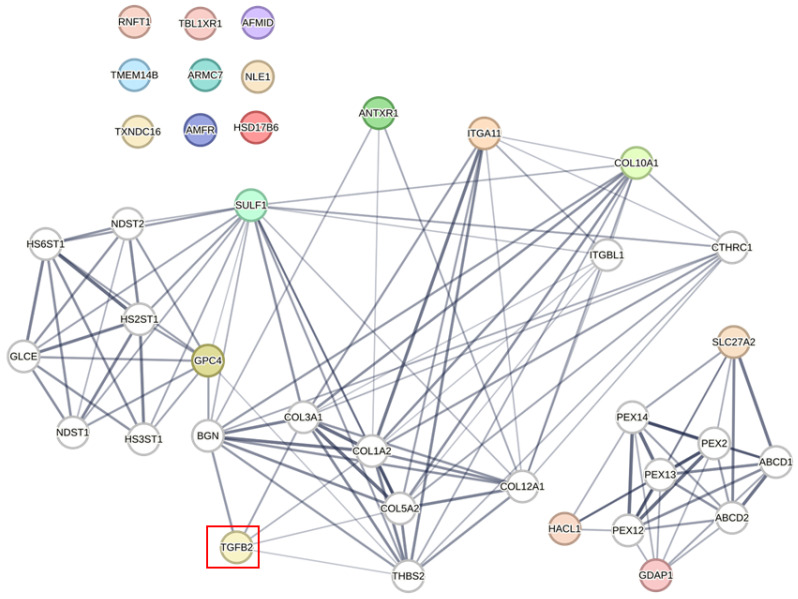
Protein–protein associations identify potential links of TGFB2 to the *TGFB2*-dependent Gene2 having prognostic impact in breast cancers. We integrated data from two screens analyzing *TGFB2*-dependent Gene2 (Figure 1 and Figure 2) to identify linker proteins, represented as white nodes, that connect TGFB2 (indicated by a red box) with the Gene2 (depicted as colored proteins) using the STRING database. The edges represent the confidence levels of each association, which are based on a combination of experimental evidence, including text mining, laboratory experiments, curated databases, co-expression analysis, neighborhood proximity, gene fusion events, and co-occurrence data. Protein associations were defined by interaction edge scores above 0.4, with line thickness indicating score thresholds at 0.4, 0.7, and 0.9. TGFB2 was associated with the network of GPB4, SULF1, ANTXR1, ITGA11, and COL10A1 via linker proteins via second shell interactors: BGN, COL3A1, COL1A2, COL5A2, and THBS2. A separate cluster of associations consisted of GDAP1, HACL, and SLC27A2. The 2 clusters represented Gene2 that exhibited differential impact on the OS derived from the multivariate Cox proportional hazards model, as distinguished by the HR values for Gene2. The cluster consisting of GPC4, SULF1, ANTXR1, ITGA11, and COL10A1 showed increases in the HR, whereas the cluster consisting of HACL, GDAP1, and SLC7A2 showed decreases in the HR. Red box highlights position of the TGFB2 protein.

**Figure 4 ijms-26-11580-f004:**
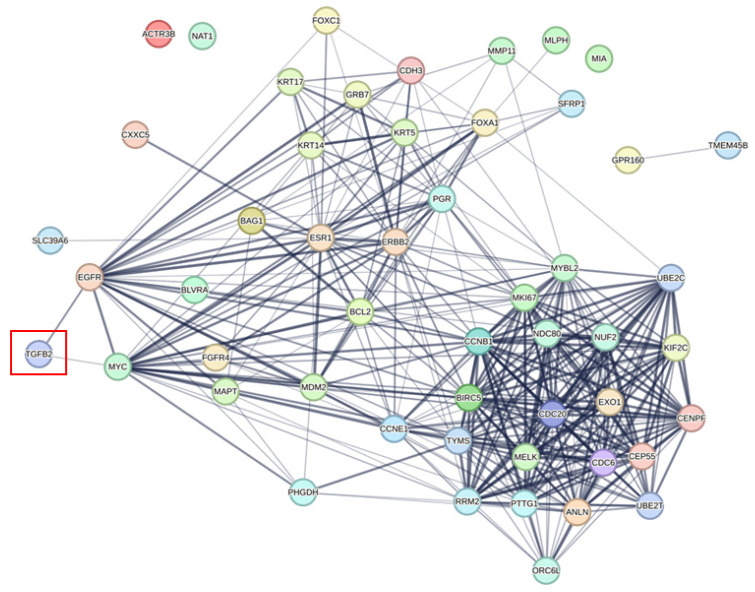
Protein–protein associations identifying potential links of TGFB2 to the PAM50 gene signature. We utilized the STRING database to identify the association of TGFB2 with the PAM50 gene signature. The edges illustrate the confidence levels of each association, which are based on a combination of experimental evidence, such as text mining, laboratory experiments, curated databases, co-expression analysis, neighborhood proximity, gene fusion events, and co-occurrence data. TGFB2 was associated with the network of the PAM50 gene signature via MYC and EGFR. Red box highlights the TGFB2 protein.

**Figure 5 ijms-26-11580-f005:**
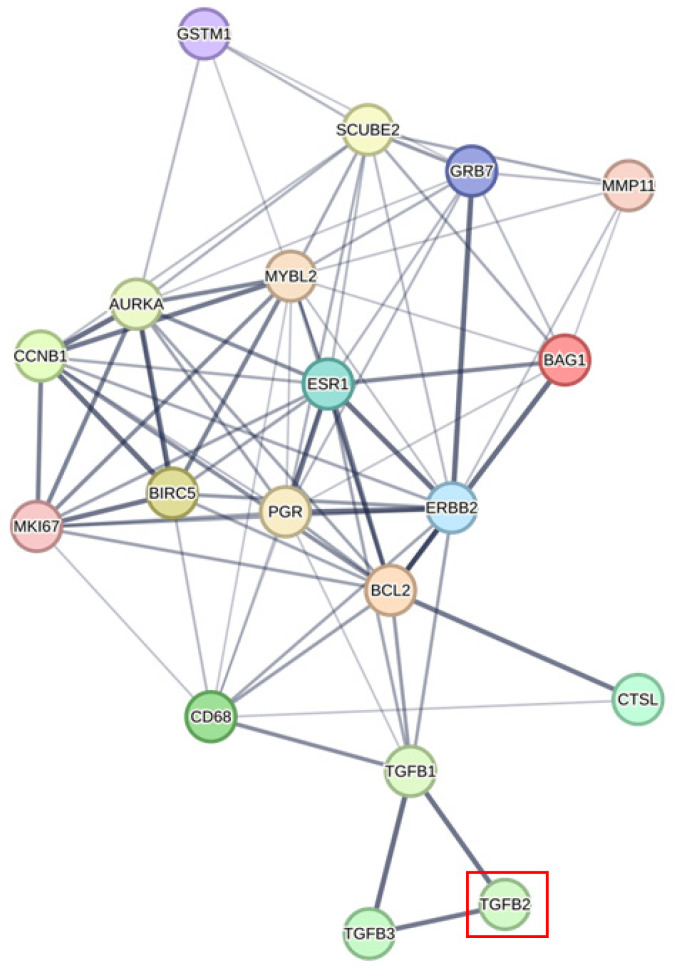
Protein–protein interactions identify potential correlations between TGFB2 and the OncotypeDx gene signature. We utilized the STRING database to identify the association of TGFB2 with the OncotypeDx gene signature. The edges represent the confidence levels of each association, which are based on a combination of experimental evidence, including text mining, laboratory experiments, curated databases, co-expression analysis, neighborhood proximity, gene fusion events, and co-occurrence data. TGFB2 was associated with the network of the OncotypeDX gene signature via TGFB1 associations with CD68, PGR, ESR1, BCL2, and ERBB2 proteins. Red box highlights the TGFB2 protein.

**Figure 6 ijms-26-11580-f006:**
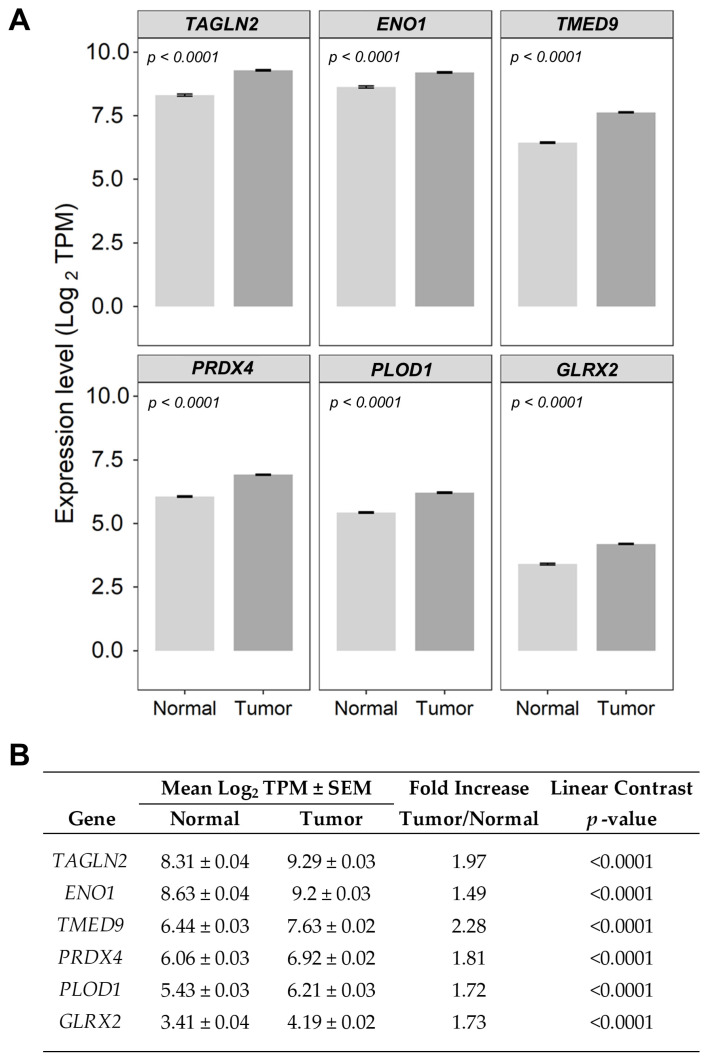
Gene expression levels for genes that showed significant prognostic impact for both the TCGA and KMplotter datasets. (**A**) The bar charts depict the expression values (mean ± SEM) of 6 significantly upregulated genes in ovarian cancer tumor tissues. They were cross-referenced in the TCGA and KMplotter datasets and ordered according to descending mRNA expression levels in the normal tissue. (**B**) The table depicts a comparison of mRNA expression values comparing normal (*n* = 179) versus tumor samples (*n* = 786 evaluable samples), ordered by expression levels in tumor tissue.

**Figure 7 ijms-26-11580-f007:**
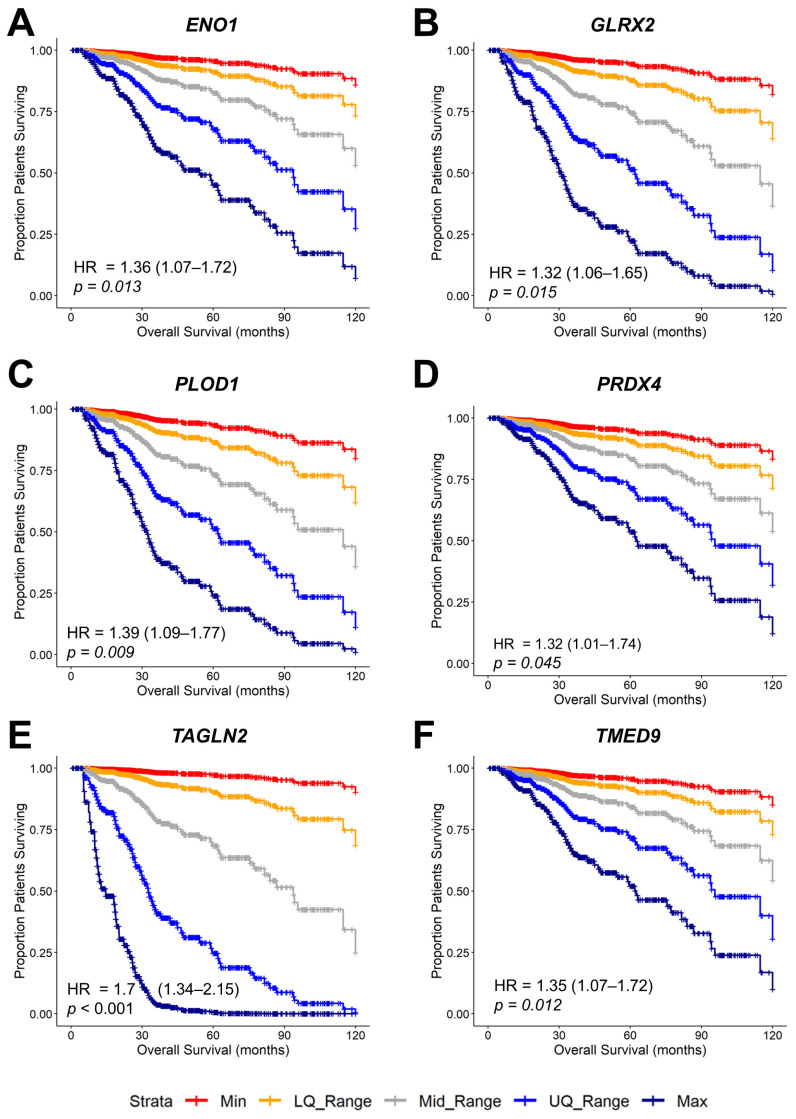
Negative prognostic impacts of Gene2 mRNA levels that were significantly upregulated in tumor tissue identified in the TCGA cohort of patients. The parameters were calculated from the Cox proportional hazards model to assess the individual effects of *ENO1* (**A**), *GLRX2* (**B**), *PLOD1* (**C**), *PRDX4* (**D**), *TAGLN2* (**E**), and *TMED9* (**F**) mRNA expression levels controlling for *TGFB2* mRNA levels, age at diagnosis, treatment (chemo-only versus all other treatments), breast cancer subtype, and the interaction between *TGFB2* and Gene2. These parameters were utilized to generate predictive OS curves (mean age = 58 yrs, *n* = 789, # death events = 59 for chemo-only treated patients, mean *TGFB2* mRNA levels, HER2+ patients) for genes exhibiting negative prognostic impact in the TCGA cohort of patients. The OS curves depict the proportion of patients surviving up to 120 months for increasing Zscores (minimum level of mRNA expression (Min), lower quartile for the range of mRNA expression (LQ Range), range of mRNA expression (Med), upper quartile for the range of mRNA expression (UQ Range) and maximum expression (Max)) of each gene (Table 1). Hazard ratios with 95% confidence intervals show a pronounced negative prognostic impact across gene expression levels. The highest level of *TAGLN2* mRNA expression exhibited the shortest median OS times (14.7 months).

**Figure 8 ijms-26-11580-f008:**
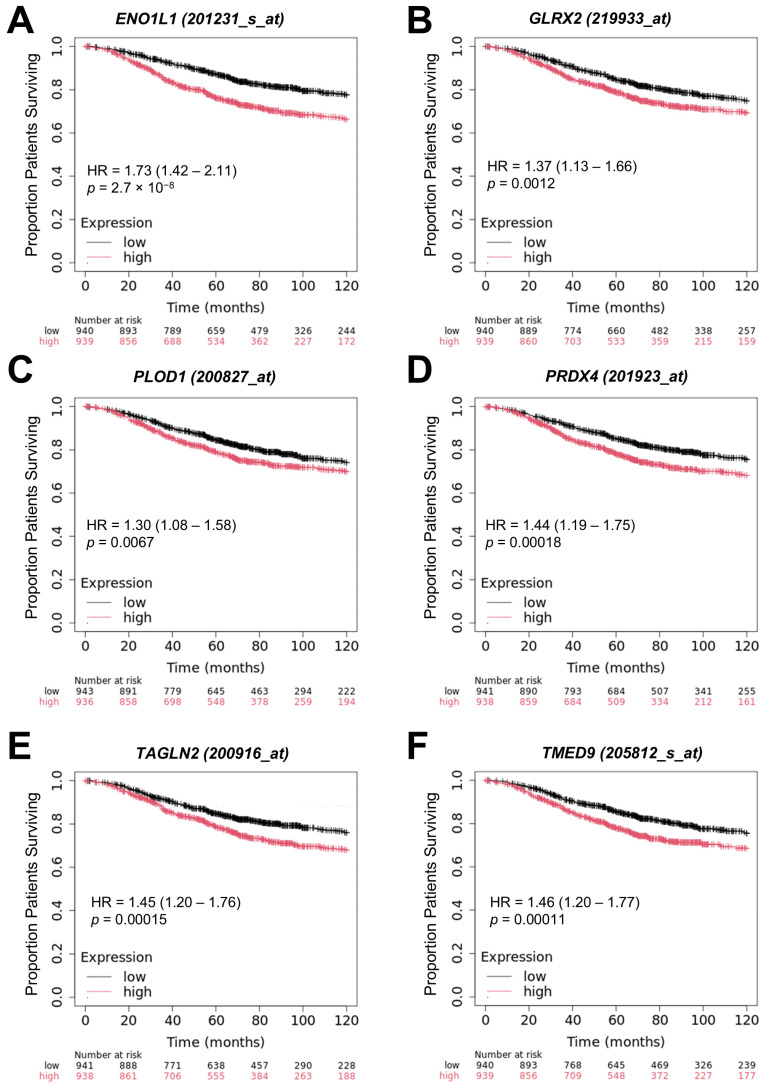
Validation of TCGA prognostic biomarkers using the independent KMplotter dataset for breast cancer patients. Six genes also significantly impacted OS when the KMplotter web tool was used to access breast cancer data from the Affymetrix dataset (*n* = 1879) and compare high versus low expression using the median cut-off for the best-selected JetSet probe set for *ENO1/ENO1L1* (**A**), *GLRX2* (**B**), *PLOD1* (**C**), *PRDX4* (**D**), *TAGLN2* (**E**), and *TMED9* (**F**).

**Table 1 ijms-26-11580-t001:** Impact on OS from increasing the mRNA levels of the negative prognostic Gene2 markers.

mRNA	Zscore	mOS (months)
*ENO1*	−1.08	>120
*ENO1*	1.27	>120
*ENO1*	3.62	>120
*ENO1*	6	93.8
*ENO1*	8.32	54.2
*GLRX2*	−1.36	>120
*GLRX2*	1.56	>120
*GLRX2*	4.47	114.8
*GLRX2*	7.39	61.9
*GLRX2*	10.3	31
*PLOD1*	−0.92	>120
*PLOD1*	1.41	>120
*PLOD1*	3.73	114.8
*PLOD1*	6.06	61.94
*PLOD1*	8.39	32.09
*PRDX4*	−1.16	>120
*PRDX4*	1.03	>120
*PRDX4*	3.22	>120
*PRDX4*	5.42	95.7
*PRDX4*	7.61	62.5
*TAGLN2*	−1.54	>120
*TAGLN2*	0.92	>120
*TAGLN2*	3.37	93.8
*TAGLN2*	5.83	32.6
*TAGLN2*	8.28	14.7
*TMED9*	−1.72	>120
*TMED9*	0.48	>120
*TMED9*	2.67	>120
*TMED9*	4.86	95.7
*TMED9*	7.06	62.5

## Data Availability

We utilized log_2_-transformed transcripts per million (TPM), summarized RNA-seq datafiles (https://toil-xena-hub.s3.us-east-1.amazonaws.com/download/TcgaTargetGtex_rsem_gene_tpm.gz (accessed on 10 November 2025); full metadata) downloaded from the UCSC Xena web platform (https://xenabrowser.net/datapages/, accessed on 25 July 2023). We analyzed clinical metadata (https://www.cbioportal.org/study/summary?id=brca_tcga_pan_can_atlas_2018; accessed on 1 February 2024) and RNA sequencing data (“data_mrna_seq_v2_rsem.txt”: batch normalized from Illumina HiSeq_RNASeqV2 using the RSEM algorithm (transcripts per million (TPM)) from 789 patients diagnosed with breast cancer to compare OS correlates with mRNA expression values.

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
