# Peer review of "Int. J. Mol. Sci.2025, 26(23), 11580;https://doi.org/10.3390/ijms262311580"

_ijms, 2025, doi:10.3390/ijms262311580_

Round 1

Reviewer 1 Report

Comments and Suggestions for Authors

References 25–27 include numerous self-citations for a single selection criterion. To the best of my knowledge, all three papers describe the same procedure. Therefore, please consider citing only the original article that introduced this method. Alternatively, it would be helpful to elaborate on the procedure in more detail within the current paper to enhance readability for the audience.

Why is it only PubMed or no other databases?

When you say abstracts, does that mean the keyword search was restricted to the abstract only or it’s the number of papers retrieved with that keyword? Please clarify

Does the chatbot validate the results? It’s a bit skeptical to rely on the AI powdered output results. I see authors have previously published similar approach, but I am wondering whether this is reliable. Did you do cross checking manually for some of them, just to validate the data you got?

It would be more appropriate if you could provide a flow diagram showing the selection criteria and the number of papers included in the analysis to aid in visualization and understanding.

In the negative prognostic biomarkers, it is not quite convincing from the heatmap and the log2 values, how the authors conclude these 6 are the top negatively correlating candidates. Could you provide more validation for this finding?

Author Response

Comment 1.

References 25–27 include numerous self-citations for a single selection criterion. To the best of my knowledge, all three papers describe the same procedure. Therefore, please consider citing only the original article that introduced this method. Alternatively, it would be helpful to elaborate on the procedure in more detail within the current paper to enhance readability for the audience.

We have taken the reviewers' suggestion to provide more details for the procedure in the supplementary methods, complete with a flowchart and a curated use of PubMed abstracts, full texts, and identification of any additional papers not deposited in PubMed, following interrogation of Perplexity AI.

“PubMed searches using the keywords: “(ARMC7 OR TMEM14B OR AMFR OR AFMID OR GDAP1) AND cancer” (171 abstracts);  “(GDAP1 OR SULF1 OR ITGA11 OR HSD17B6 OR COL10A1) AND cancer” (410 abstracts); “TGF-beta AND Breast” (3667 abstracts); “Gene AND TCGA AND prognosis AND breast” (2273 abstracts) (https://pubmed.ncbi.nlm.nih.gov/ accessed 18th  December 2024) were downloaded as text documents for processing using the Oncotelic Chatbot technologies. We also utilized Perplexity AI chatbot to identify articles (https://www.perplexity.ai/, accessed 27th October, 2025) to identify articles not deposited in PubMed. Detailed methods are described in the supplementary methods.   Chatbot technology enabled interactive keyword refinement for PubMed keyword searches by conducting question-and-answer sessions to refine the references; however, it was not utilized for systematic review, result validation, hypothesis formulation, or manuscript writing. All biomarker discoveries in the manuscript resulted from bioinformatic analyses and identification of these genes were used to interrogate PubMed and Perplexity AI tools to contextualize the literature for these genes.” (Lines 714-728)

Comment 2.

Why is it only PubMed or no other databases?

PubMed provides access to its database via an API that can be integrated into the workflow.  We have now detailed in the supplementary methods how Perplexity AI was utilized to identify additional manuscripts for manual curation.

Comment 3.

When you say abstracts, does that mean the keyword search was restricted to the abstract only or it’s the number of papers retrieved with that keyword? Please clarify

We selected “abstracts” in the PubMed downloading tool that only retrieved the abstracts. Full text papers were downloading for manual curation following the question-answer sessions with the Oncotelic Chatbot and Perplexity AI tools.

Comment 4

Does the chatbot validate the results? It’s a bit skeptical to rely on the AI powdered output results. I see authors have previously published similar approach, but I am wondering whether this is reliable. Did you do cross checking manually for some of them, just to validate the data you got?

The chatbot was used solely to identify published literature for contextualizing the biomarkers. The biomarker discovery and validation protocols, utilizing TCGA and KMplotter datasets, were all bioinformatics-driven.  The Cox proportional hazards model identified TGFB2-dependent and independent prognostic markers.  The Chatbots played no part in the validation of the results.  Following the interrogation of the chatbots through question-answer sessions, full-text papers were obtained to write the introduction and discussion sections of the paper, based on a manual reading of the papers.  We found that relying solely on AI-powered outputs was not reliable.

Comment 5

It would be more appropriate if you could provide a flow diagram showing the selection criteria and the number of papers included in the analysis to aid in visualization and understanding.

We have provided a flow chart in the supplementary methods

Comment 6

In the negative prognostic biomarkers, it is not quite convincing from the heatmap and the log2 values, how the authors conclude these 6 are the top negatively correlating candidates. Could you provide more validation for this finding?

We did not use tumor versus normal comparisons to select the biomarkers; as such, we have now moved the heatmap to the supplementary section to avoid distraction. The prognostic markers were selected from filtering the genes based on the p-value for the interaction term in the Cox proportional hazards model.  Tumor versus normal comparisons were used to determine whether the prognostic markers were upregulated in the tumors. The final 6 prognostic markers were cross-validated in both TCGA and KMplotter datasets.

The results section details the filtering procedure (lines 252-273):

“We next performed multivariate analyses using the Cox proportional hazards model to assess the individual effects of TGFB2 and Gene2 marker mRNA expression levels, identifying marker genes that exhibited an increase in HR, independent of TGFB2 mRNA expression, on OS. This analysis was controlled for age at diagnosis, breast cancer subtype, the confounding effect of treatment (Chemotherapy only versus all other treatments), and the interaction between TGFB2 and Gene2. Out of the 15,947 marker genes screened, 1286 exhibited increases in HR for Gene 2 (p<0.05), of which 118 genes exhibited increases in HR for Gene 2 not impacted by TGFB2 expression (p-value for TGFB2 > 0.1 AND p-value for TGFB2 by Gene 2 interaction > 0.1 AND Chemotherapy only parameter > 1.5 standard deviation units).  We compared the mRNA expression in tumor tissues of 118 Gene2 markers relative to normal tissues, resulting in 41 genes significantly upregulated in tumor tissues for 786 evaluable patients (Figure S2, Table S4; using the filter: p <0.0001, fold-increase > 1, log2 (TPM) expression in tumor tissue > 2).

We cross-referenced the 41 genes identified in the TCGA dataset with those reported in the KMplotter database to narrow down the potential list of prognostic markers using more stringent criteria, comparing only median cut-off values for high versus low-expressing patient subsets.  This analysis identified 6 significantly upregulated genes in breast cancer tumor tissues cross-referenced in both TCGA and KMplotter datasets (Figure 6). A significant increase in mRNA expression levels was observed in tumor tissue compared to normal tissue for the following genes: TAGLN2 (2-fold, p < 0.0001), ENO1 (1.49-fold, p < 0.0001), GLRX2 (1.73-fold, p < 0.0001), PLOD1 (1.72-fold, p < 0.0001), PRDX4 (1.81-fold, p < 0.0001), and TMED9 (2.28-fold, p < 0.0001) (Figure 6B).”

Reviewer 2 Report

Comments and Suggestions for Authors

Overall the manuscript is well written and well formatted. There are few concerns before it can be published in this journal.

Minor comments:

  1. Line 22: The phrase "worse outcomes" can be replaced with a better word/phrase.
  2. Line 150: What does the authors mean by "negative prognostic biomarkers"?
  3. Line 180: The authors used "two-way ANOVA model" for DEG analysis. Why other differential expression method like DESeq2 was not used?
  4. In section 2.3 there are many occurrences of "Gene 2" without describing what it is. Please mention at the first occurrence what it means (e.g. marker gene) to remove confusion.
  5. Line 225: What do the authors mean by "survival proportion"?
  6. Line 253: A fold change of  877.9 seems absurd. Is it a typing mistake? Please look into it.
  7. The quality of Figure 2 should be improved.
  8. The authors have used subheadings for Discussion section. Is it really required?

Author Response

Minor comments:

Line 22: The phrase "worse outcomes" can be replaced with a better word/phrase.

We have removed the simple summary section. We did clarify a similar statement in the results section (Lines 252-254)

“We next performed multivariate analyses using the Cox proportional hazards model to assess the individual effects of TGFB2 and Gene2 marker mRNA expression levels, identifying marker genes that exhibited an increase in HR, independent of TGFB2 mRNA expression, on OS.”

Line 150: What does the authors mean by "negative prognostic biomarkers"?

We have now rephrased the sentence (Lines 890-891)

“exhibited a negative correlation between mRNA expression and OS across all expression levels in the breast cancer patient, validated with an independent data set from the KMplotter database.  ”

Line 180: The authors used "two-way ANOVA model" for DEG analysis. Why other differential expression method like DESeq2 was not used?

DESeq2 was developed to identify differential expressed genes directly from the raw count data, especially for distributions skewed strongly to low or zero counts.  For the purposes of normal versus tumor analysis, the data sets would have to be combined from two different data sources: TCGA and GTEx, and that data would need to be further processed using batch corrections.  We instead used the pre-processed data from the Xena web portal and used an analogous technique using RSEM algorithms for normalized count data with appropriate data transformations to enable the use of ANOVA techniques.  We have now detailed this in the methods section (lines 741-759).

“The gene expression measure in the Xena database, especially for TCGA cohorts, is normalized, log2-transformed RSEM values represented as log2(x + 1), where x is the normalized expression value estimated by the RNA-Seq Expectation-Maximization (RSEM) method. This metric is designed to facilitate cross-sample comparisons by stabilizing variance and normalizing variations in sequencing depth. We applied ANOVA methods to determine differential gene expression values such as RSEM normalization and transformations for the following features: 1. Variance stabilization to make the data more normally distributed, satisfying key ANOVA assumptions;  2). Variance modeling by quantifying uncertainty and variance in expression estimates by incorporating read mapping ambiguity and fragment length distributions during expectation maximization; and 3. Handling heteroscedasticity: by modelling the variance-mean dependency explicitly. Log-transforming RSEM values reduces heteroscedasticity (unequal variance), thereby improving the validity of ANOVA or related variance component tests on expression data. There are related methods (DESeq2/edgeR) that leverage count reads directly for gene-level comparisons.  The Xena platform provides batch-corrected RSEM-normalized and transformed expression values from two different datasets, including GTEx normal samples, TCGA tumor samples, and adjacent normal samples, which enables robust comparisons between normal and tumor samples.”

In section 2.3 there are many occurrences of "Gene 2" without describing what it is. Please mention at the first occurrence what it means (e.g. marker gene) to remove confusion.

We have now implemented this suggestion .

“TGFB2 mRNA, an mRNA product of a marker gene (Gene2) expression” (Line 116)

Line 225: What do the authors mean by "survival proportion"?

We have now clarified this sentence (Line 820)

“we simulated the proportion patients surviving (number at risk at follow up time points relative to number at risk at diagnosis)”

Line 253: A fold change of  877.9 seems absurd. Is it a typing mistake? Please look into it.

This high fold change was a result of very low level of expression in normal tissue of less than 1 trancript per 10 million in normal tissue.  We have revised the statement on line 144:

“Expression of COL10A1 mRNA exhibited a significant (p < 0.0001) 877.9-fold change in tumor compared to normal tissue; this high value for fold-change was a result of the very low expression of COL10A1 in normal tissue (Mean ± SEM = −3.92 ± 0.16 log2TPM).”

The quality of Figure 2 should be improved.

We have now included a high-resolution tiff image in the manuscript.

The authors have used subheadings for Discussion section. Is it really required?

We agree that the sub-headings were not required and have removed these headings.

Reviewer 3 Report

Comments and Suggestions for Authors

In this manuscript, the authors conduct a bioinformatic analysis to identify prognostic biomarkers in breast cancer, classifying them as either TGFB2-dependent or -independent. Using a multivariate Cox proportional hazards model with an interaction term applied to public datasets (TCGA and KMplotter), they report seven potential positive and six negative prognostic markers. The study addresses a clinically relevant question with a methodologically sound approach. However, in my view, several limitations should be addressed before publication.

1.The authors state that they compared “patients who received and performed worse in chemotherapy-only therapies.” This aspect of the analysis is methodologically critical but currently lacks clarity. The term “performed worse” requires precise operational definition—whether it refers to overall survival, progression-free survival, tumor response rate, or another clinical endpoint. If the stratification was based on survival outcomes and the same outcome is used in the prognostic model, there is a potential risk of circularity or selection bias. The authors should provide a detailed explanation of the stratification criteria and methodological justification to address this concern.

2.The Results section presents extensive statistical output (e.g., hazard ratios, p-values, confidence intervals). While thorough, the density of data can obscure the key findings. It is recommended that the authors streamline the narrative to improve readability and highlight the central results, without compromising scientific rigor.

3.A key innovation of this study is the use of an interaction term in the Cox model to define TGFB2-dependent prognostic markers. However, the manuscript does not fully explain the statistical rationale under which a significant interaction term translates to “dependency.” Expanded discussion in the Methods or Results section—clarifying both the statistical reasoning and biological interpretation—would strengthen the manuscript.

4.The terms “prognostic” and “predictive” are used interchangeably in places, though they convey distinct meanings. Prognostic markers reflect disease outcomes irrespective of treatment, while predictive markers signal response to a specific therapy. Since this study analyzes overall survival in a retrospective cohort, the findings are primarily prognostic. Claims regarding predictive value should either be tempered and explicitly qualified, or the terminology should be revised throughout for greater accuracy.

Comments on the Quality of English Language

The English language quality of the manuscript is generally good and the scientific content is clearly communicated. However, some sections, particularly the description of methodological groupings and the differentiation between key statistical terms, could be clarified to enhance precision and avoid potential misinterpretation.

Author Response

1.The authors state that they compared “patients who received and performed worse in chemotherapy-only therapies.” This aspect of the analysis is methodologically critical but currently lacks clarity. The term “performed worse” requires precise operational definition—whether it refers to overall survival, progression-free survival, tumor response rate, or another clinical endpoint. If the stratification was based on survival outcomes and the same outcome is used in the prognostic model, there is a potential risk of circularity or selection bias. The authors should provide a detailed explanation of the stratification criteria and methodological justification to address this concern.

We thank the reviewer for pointing out this concern.  We used the entire TCGA cohort for the Cox proportional hazards model.  The chemotherapy was a control variable in the model, not a selection criteria for analyzing the patient cohort.  We filtered the prognostic genes that exhibited an increase in hazard ratio for chemo-only patients.  We have now added the following clarifications:

“We implemented a multivariate Cox proportional hazards model to directly compare hazard ratio (HR) calculations for TGFB2 mRNA, a marker gene expression, including an interaction term of TGFB2 by marker gene expression, while controlling for age at diagnosis, breast cancer subtypes, and comparing patients who received chemotherapy-only therapies, and filtered genes that reported HR greater than 1 in the multivariate model. The interaction term in the model enabled the identification of TGFB2/marker gene combinations that result in synergistic improvements for breast cancer patients. This resulted in TGFB2-dependent positive prognostic markers that could be potentially used as inclusion criteria in biomarker-guided clinical trial designs.” (Lines 118-123)

2.The Results section presents extensive statistical output (e.g., hazard ratios, p-values, confidence intervals). While thorough, the density of data can obscure the key findings. It is recommended that the authors streamline the narrative to improve readability and highlight the central results, without compromising scientific rigor.

We thank the review for this suggestion to improve readability.  We have added tables to figures 1, 2 and 6, and included a new table 1 that illustrate the statistics, so to reduce the reporting in the results text.

3.A key innovation of this study is the use of an interaction term in the Cox model to define TGFB2-dependent prognostic markers. However, the manuscript does not fully explain the statistical rationale under which a significant interaction term translates to “dependency.” Expanded discussion in the Methods or Results section—clarifying both the statistical reasoning and biological interpretation—would strengthen the manuscript.

We have now clarified the use of the statistical interaction term and TGFB2-dependency.

“Multivariate analyses utilized the Cox proportional hazards model to assess the individual effects of TGFB2 and marker gene (Gene2) mRNA expression levels (screened 15,947 genes) on overall survival (OS) to calculate hazard HR. This analysis was controlled for age at breast cancer subtype, diagnosis, treatment (Chemotherapy only versus all other treatments), and the interaction between TGFB2 and Gene2 mRNA. The inclusion of a statistical interaction term was an important consideration in our analysis design. Gene expression of TGFB2 mRNA and Gene2 mRNA were the main effects of the model. The interaction coefficients from the interaction term in the model represents the additional effect of the interaction term at a specific expression level of TGFB2 mRNA on Gene2 expression levels; the interaction term determines how much more the effect of TGFB2 mRNA levels impacts the effect of Gene2 mRNA expression.” (Lines 786-796).

“Gene expression values were correlated to OS outcomes investigating the impact of mRNA expression of TGFB2 (median cut-off for high and low levels of expression), further stratified into four groups based on gene expression levels of Gene2 mRNA expression levels (median cut-off for high and low levels) in these patients. We defined TGFB2-dependency by visualizing the significant the interaction effect from the multivariate Cox-proportional hazards models using Kaplan-Meier analysis with 50th percentile cut-offs for TGFB2 and Gene2. This analysis was implemented to show that when comparing high versus low levels of Gene2 expression the impact on OS was different comparing high and low levels of TGFB2 mRNA expression or when comparing high versus low levels of TGFB2 mRNA the OS curves were differentially impacted at low and high levels of Gene2. We identified TGFB2/Gene2 combinations that exhibited significant curve separation between the two arms of the four survival curves (p < 0.05), whereby high levels of TGFB2 and maker gene mRNA expression resulted in the most improved OS outcomes.” (Lines 835-848)

4.The terms “prognostic” and “predictive” are used interchangeably in places, though they convey distinct meanings. Prognostic markers reflect disease outcomes irrespective of treatment, while predictive markers signal response to a specific therapy. Since this study analyzes overall survival in a retrospective cohort, the findings are primarily prognostic. Claims regarding predictive value should either be tempered and explicitly qualified, or the terminology should be revised throughout for greater accuracy.

We thank the reviewer this comment and we have now revised the use of the terminology to reflect that the findings were primarily prognostic.

Reviewer 4 Report

Comments and Suggestions for Authors

The manuscript is thoughtfully designed and offers valuable insights, as it links candidate biomarker genes to the TGFB2 gene in breast cancers using bioinformatic data. Nevertheless, I would like to offer some constructive suggestions for improvement:

  • First, please carefully review all abbreviations used throughout the manuscript. It is preferable to avoid using the long forms of abbreviations repeatedly (e.g., OS, HR) and ensure consistency throughout the text. This will improve readability and clarity.
  • The term "mo0del" on line 219 appears to be a typographical error and should be corrected.
  • All gene names should be italicized consistently throughout the manuscript, as per standard scientific conventions.
  • The resolution of Figure 2 could be enhanced to ensure that all details are clearly visible, facilitating better interpretation by readers.
  • The sentence on lines 443-444, "SULF1 mRNA expression has been implicated in the prognosis of various types of cancer for both positive and negative prognostic impact," could benefit from further clarification. Specifically, it would be helpful to briefly explain why SULF1 expression varies across different cancer types, supported by relevant literature. This would strengthen the discussion and provide readers with a clearer understanding of the gene’s context.
  • Latin terms such as in vivo and in vitro should be italicized throughout the manuscript. A careful review to ensure consistency would be beneficial.
  • In the Conclusions section (lines 649-651), the correlation between the genes "GDAP1, TBL1XR1, RNFT1, HACL1, SLC27A2, NLE1, and TXNDC16" and TGFB2 is noted. To avoid leaving this observation unsubstantiated, it would be valuable to discuss potential causal or mechanistic relationships between these genes and TGFB2 in light of the existing literature. As a suggestion, conducting a STRING analysis could help illustrate these relationships through an in silico network map, supporting a more thorough discussion.
  • The statement on lines 658-659, "suggest that either these genes could be presented as targets for therapy," might come across as somewhat assertive. It would be advisable to present this suggestion more cautiously unless supported by clinical data. Similarly, the phrase "should be excluded from a clinical trial" is a strong claim and could be softened to reflect a more tentative interpretation.
  • In the Discussion section, beyond comparing findings with the literature, it would strengthen the manuscript to provide an evaluative sentence for each gene discussed, conveying the authors’ scientific interpretations and helping readers understand the broader significance of the findings.

Overall, these refinements would enhance the clarity, readability, and scientific rigor of the manuscript, while maintaining its valuable contributions to the field.

Author Response

Comment 1

First, please carefully review all abbreviations used throughout the manuscript. It is preferable to avoid using the long forms of abbreviations repeatedly (e.g., OS, HR) and ensure consistency throughout the text. This will improve readability and clarity.

We thank the reviewer for this comment and made the revisions accordingly.

Comment 2

The term "mo0del" on line 219 appears to be a typographical error and should be corrected.

We have corrected the typo.

Comment 3

All gene names should be italicized consistently throughout the manuscript, as per standard scientific conventions.

We thank the reviewer for their suggestion.  We have revised the definition of Gene2 to make clear that when referring to mRNA gene product, or protein name, that the Hugo gene symbol remained in capitol letters.

“TGFB2 mRNA, an mRNA product of a marker gene (Gene2) expression” (116)

Comment 4

The resolution of Figure 2 could be enhanced to ensure that all details are clearly visible, facilitating better interpretation by readers.

We have now included a high-resolution tiff image in the manuscript.

Comment 5

The sentence on lines 443-444, "SULF1 mRNA expression has been implicated in the prognosis of various types of cancer for both positive and negative prognostic impact," could benefit from further clarification. Specifically, it would be helpful to briefly explain why SULF1 expression varies across different cancer types, supported by relevant literature. This would strengthen the discussion and provide readers with a clearer understanding of the gene’s context.

We have updated the discussion section to clarify the original statement:

Comment 6

Latin terms such as in vivo and in vitro should be italicized throughout the manuscript. A careful review to ensure consistency would be beneficial.

We have revised the manuscript according to the reviewers suggestion.

“Heparin sulfate 6-O-endosulfatases 1 (SULF1) mRNA exhibited a significant increase in breast tumor tissue. In our multivariate Cox proportional hazards model, increasing levels of SULF1 increased the HR in breast cancer patients (22% increase for one standard deviation increase in gene expression), in contrast to the positive prognostic impact of TGFB2 mRNA levels. Network analysis SULF1 exhibited a second-order connection to TGFB2 via BGN, COL3A1, COL1A2, COL5A2, and THBS2. 

SULF1 is a member of the sulfatase family, which regulates the sulfation of Heparan sulfate proteoglycans and has been implicated in the prognosis of various types of cancer for both positive and negative prognostic impact. SULF1 mRNA expression was associated with a poor prognosis in lung adenocarcinoma[25]. A study that employed quantitative real-time polymerase chain reaction to measure SULF1 mRNA expression from 54 non-small cell lung cancer (NSCL) patients showed patients expressing high levels of SULF1 mRNA exhibited shorter mOS times. Furthermore, knockdown of SULF1 suppressed key malignant behaviors in NSCLC cell lines, including NCI-H1299 (CRL-5803) and HCC827 (CRL-2868), by reducing cell proliferation, migration, invasion, and epithelial-mesenchymal transition (EMT) through inhibition of the EGFR/MAPK signaling pathway [26].  In hepatocellular carcinoma (HCC), SULF1 promotes TGF-β-induced gene expression and epithelial-mesenchymal transition [27]. In vitro studies demonstrated that forced SULF1 expression in HCC cell lines (Hep3B, PLC/PRF/5) resulted in increased SMAD2/3 phosphorylation following stimulation with TGF-β1. Conversely, SULF1 knockdown in cell lines with high endogenous levels (SNU182, SNU475) reduced TGF-β signaling, migration, and invasiveness [27]. Furthermore, both immunohistochemistry and Western blotting identified elevated SULF1 expression and increased phosphorylation of SMAD2/3 (TGF-β pathway activation) in tumor and peritumoral tissues from transgenic Sulf1-Tg mice compared to WT [27] . SULF1 expression facilitated the upregulation of the hallmarks of EMT mesenchymal markers: N-cadherin, vimentin, and αSMA [27]. Notably, these observations were based on the activation of TGF-β1 expression, whereas our prognostic model suggested that the TGFB2 mRNA isomer drove a positive prognostic impact in breast cancers, indicating the need for a careful and nuanced examination of the TGFB ligand isoforms in tumor progression. In gastric cancer, [28] Fang et al 2024, reported that SULF1, produced by cancer-associated fibroblasts (CAFs), facilitated metastasis and resistance to cisplatin treatment via TGF-β1 activation of TGFBR3-mediated signaling. Pan cancer analysis revealed that SULF1 mRNA was upregulated more than 2-fold in 16 out of the 32 cancers in the TCGA dataset [29]. Analysis of a single-cell RNA-seq dataset of Head and Neck Squamous Cell Carcinoma (HNSC) showed the highest positivity of SULF1 in fibroblasts [29]. In pancreatic cancer, high SULF1 expression has been associated with later T, N, and TNM stages, higher CA19-9 levels, smaller tumor size, and poorer prognosis [30].

Tumor suppressor role for SULF1 being an extracellular sulfatase that removes 6-O-sulfate groups from heparan sulfate (HS) chains, thereby reducing the activity of HS-binding growth factors such as FGF2, VEGF, amphiregulin, HB-EGF, and HGF [31] .By removing these sulfate groups, SULF1 decreases growth factor presence at the cell surface and restricts receptor interaction, ultimately suppressing pro-tumorigenic signaling [31].

In breast cancers under hypoxic conditions, SULF1 is downregulated by HIF-1α, promoting cancer cell migration and invasion, in aggressive breast cancer cell lines [32]. In this study, Kaplan Meier survival analysis showed that tumors with high SULF1 mRNA expression exhibited longer disease-free survival OS compared to patients whose tumors had low levels of SULF1 mRNA expression [32]. Ovarian cancer cell lines and primary tumors lacking SULF1 mRNA expression showed dense DNA methylation in 12 CpG sites within exon 1A and increased histone H3 methylation around the SULF1 gene promoter, leading to transcriptional repression [33]. Downregulation of SULF1 mRNA in ovarian cancer cells, achieved via siRNA, reduces sensitivity to cisplatin-induced cytotoxicity, indicating that loss of SULF1 mRNA promotes chemoresistance. Furthermore, loss of SULF1 leads to decreased expression of the pro-apoptotic protein Bim via increased ERK signaling, promoting tumor cell survival and resistance to chemotherapy [34].

Taken together, the impact of SULF1 in the tumor microenvironment is highly context-dependent on the tumor type, expression in stromal cells, the mechanism of SULF1 gene activation, and hypoxia. SULF1 interacts with the TGFB-beta pathway via TGF-beta 1. Our analysis high levels of SULF1 are overexpressed in tumors and has a negative prognostic impact in the context of low levels of a different TGF-beta isomer, TGFB2, suggested by the significant statistical interaction term calculated from the multivariate Cox proportional hazards term.  The network analysis showed no direct mechanistic connection of TGFB2 and SULF1, but they are components of a highly interconnected set of associations. Further work will need to elucidate the expression of TGFB1, TGFB2, TGF-beta receptors, and SULF1 using single-cell RNA-seq studies for the distribution of expression in the cellular compartments in tumors.” (Lines 354-420)

Comment 7

In the Conclusions section (lines 649-651), the correlation between the genes "GDAP1, TBL1XR1, RNFT1, HACL1, SLC27A2, NLE1, and TXNDC16" and TGFB2 is noted. To avoid leaving this observation unsubstantiated, it would be valuable to discuss potential causal or mechanistic relationships between these genes and TGFB2 in light of the existing literature. As a suggestion, conducting a STRING analysis could help illustrate these relationships through an in silico network map, supporting a more thorough discussion.”

We thank the reviewer for this suggestion and we have performed STRING analysis depicted in the new figure 3 and further discussed the results.

“Investigation of the TGFB2-dependent gene signatures for their protein-protein associations from the two screens in our study demonstrated that TGFB2 was associated with a network consisting of GPC4, SULF1, ANTXR1, ITGA11, and COL10A1 via linker proteins: BGN, COL3A1, COL1A2, COL5A2, and THBS2. Studies have demonstrated that TGF-beta induces BGN expression in pancreatic cells through the activation of MKK6-p38 MAPK signaling downstream of Smad signaling, offering an insight into the regulation of BGN observed in fibrosis and desmoplasia associated with inflammatory responses [68]. Furthermore, the TGF-beta/ALK5 effect on p38 activation and BGN expression was also impacted by overexpression of GADD45beta alone in PANC-1 and osteosarcoma MG-63 cells [69]. Research directed towards identifying direct Smad targets in dermal fibroblasts following TGF-beta stimulation, requiring demonstration of a rapid increase in mRNA levels, TGF-beta-induced promoter activity blocked by dominant-negative Smad3 and Smad7 vectors; and no promoter transactivation in Smad3(-/-) fibroblasts found COL3A1, as a TGF-beta/Smad3 target [70]. TGF-β2 treatment elevates COL1A2 expression in human epithelial SRA01/04 cells whereby COL1A2 knockdown inhibited TGF-β2-induced SRA01/04 cell proliferation, migration, invasion and EMT [71]. COL5A2 expression was shown to be elevated in human osteosarcoma cells, and the downregulation of COL5A2 affected the TGF-β and Wnt/β-Catenin signaling pathways [72]. In this study, bioinformatics analysis was employed to assess the prognostic significance of COL5A2 in osteosarcoma. The findings indicated that COL5A2 inhibits osteosarcoma invasion and metastasis by suppressing both the TGF-β and Wnt/β-Catenin signaling pathways [72]. THBS2 expression in pancreatic cancer is mainly present in the stroma and is linked to tumor progression and poor prognosis. RNA in situ hybridization revealed that CAFs, not tumor cells, express THBS2, with levels increasing as the disease progresses in mouse models [72]. TGF-β1 from cancer cells activated CAFs to produce THBS2 through the p-Smad2/3 pathway, suggesting an important role for TGF-β association with THBS2 in cancer progression [73].” (Lines 508-533)

Comment 8

The statement on lines 658-659, "suggest that either these genes could be presented as targets for therapy," might come across as somewhat assertive. It would be advisable to present this suggestion more cautiously unless supported by clinical data. Similarly, the phrase "should be excluded from a clinical trial" is a strong claim and could be softened to reflect a more tentative interpretation.

We thank the reviewer’s comment and have revised the statements to be less assertive.

“A multivariate analysis revealed that when controlling TGFB2 expression, we identified six genes (ENO1, GLRX2, PLOD1, PRDX4, TAGLN2, and TMED9) that exhibited a negative correlation between mRNA expression and OS across all expression levels in the breast cancer patient, validated with an independent data set from the KMplotter database.  Patients expressing low levels of these 6 genes would benefit from standard therapies as they showed improved survival outcomes at low mRNA levels of expression. Five of these negative prognostic markers are druggable (ENO1, PLOD1, PRDX4, TAGLN2, and TMED9). The increased HR for patients with high levels of expression of these marker genes suggests that these five genes could be presented as future development for targeted therapies” (Lines 887-896)

Comment 9

In the Discussion section, beyond comparing findings with the literature, it would strengthen the manuscript to provide an evaluative sentence for each gene discussed, conveying the authors’ scientific interpretations and helping readers understand the broader significance of the findings.

We thank the reviewer for this comment and have revised the manuscript accordingly.

“Taken together, the impact of SULF1 in the tumor microenvironment is highly context-dependent on the tumor type, expression in stromal cells, the mechanism of SULF1 gene activation, and hypoxia. SULF1 interacts with the TGFB-beta pathway via TGF-beta 1. Our analysis high levels of SULF1 are overexpressed in tumors and has a negative prognostic impact in the context of low levels of a different TGF-beta isomer, TGFB2, suggested by the significant statistical interaction term calculated from the multivariate Cox proportional hazards term.  The network analysis showed no direct mechanistic connection of TGFB2 and SULF1, but they are components of a highly interconnected set of associations. Further work will need to elucidate the expression of TGFB1, TGFB2, TGF-beta receptors, and SULF1 using single-cell RNA-seq studies for the distribution of expression in the cellular compartments in tumors.” (410-420)

“suggesting that breast cancer patients with high levels of both TGFB2 and GDAP1 expression displayed significantly improved OS outcomes.  Therefore, patients expressing high mRNA levels of these two genes would be most susceptible to standard therapies.” (Lines 468-469)

“Our studies also suggested that the prognostic effect of SLC27A2 was context-dependent on TGFB2 mRNA levels, whereby high levels of SLC27A2 and TGFB2 mRNA levels showed improved OS outcomes, identifying patients who could benefit from standard therapies.” (Lines 482-484).

“Our results suggest that ENO1 is targetable for treated patients with high levels of ENO1 mRNA.” (Line598)

“Our results suggest that breast cancer patients with low levels of GLXR2 expression will exhibit improved OS outcomes with standard therapies.” (Line 613)

“Our study suggests that patients expressing high levels of PLOD1 can be targeted for therapies against Prolyl 4-hydroxylase alpha subunits.” (Lines 633)

“Our results suggest that breast cancer patients with high levels of PRDX4 may be targeted using compounds such as Con A.” (Lines 653)

“Our results suggest that breast cancer patients with high levels of TMED mRNA are potentially susceptible to inhibiting TMED.” (Line 691)

“Although no clinical trials are testing these compounds, inhibition of TAGLN2 may be pursued as a therapeutic option for breast cancer patients expressing high levels of this gene.” (Line 680)

Reviewer 5 Report

Comments and Suggestions for Authors

This manuscript offers a bioinformatics-based analysis aimed at identifying prognostic biomarkers in breast cancer, emphasizing the role of TGFB2 mRNA expression. Overall, this is a good preliminary study but needs revisions for clarity, reproducibility, and wider impact.

  1. In “simple summary” and “introduction”, the authors stated that these negative markers provide a list of potential targets for therapies. However, the results focus more on their prognostic potential. Authors need to either expand on the therapeutic potential by mentioning ongoing clinical trials or revise the statement.
  2. In the introduction, the paragraph beginning on line 71 appears unrelated. It is recommended to move it to the discussion if relevant, which will help make the introduction more concise.
  3. The last paragraph of the introduction repeats information from other sections and can be summarized more concisely.
  4. AI-Powered PubMed Search is innovative but underdeveloped. The Oncotelic Chatbot is referenced, yet no details on its architecture, query refinement, or bias mitigation are provided. Authors need to provide more details in the method section or supplementary materials.
  5. Authors are advised to include some of the data presented in supplementary tables as main figures or as a table in the main text, particularly the HR of selected genes.
  6. The quality of Figure 2 indeed needs to be improved.
  7. Authors are advised to mention the HR of genes only once, as they were repeated multiple times.
  8. Add P values for Figure 4
  9. To elevate significance, discuss how these markers inform specific therapies such as TGFB2 inhibitors or integrate with current standards like PAM50/Oncotype DX.
  10. 7 positive markers mentioned in lines 19–20 conflict with the 11 markers in lines 248–249; please clarify.
  11. COL10A1's 877.9-fold change seems extreme; verify log2TPM calculations. A figure showing the full heatmap would aid interpretation.
  12. Authors need to refine the discussion and avoid repeating results data in this section, as well as avoid mentioning tables and figures.
  13. Authors are requested to expand the discussion to address limitations.
  14. Authors are requested to explain how these markers might stratify patients for anti-TGFB2 trials.
  15. Authors are requested to mention Future directions, such as scRNA-seq integration.
  16. Some authors with a conflict of interest are involved in designing the study. The particular role of the author needs more details.
  17. There are some typos and grammatical errors. Authors are encouraged to revise the manuscript grammatically.
  18. Authors need to double-check their manuscript for plagiarism and avoid using parts of other published papers, particularly in section 2.2 and Lin 89-99

Author Response

Comment 1

In “simple summary” and “introduction”, the authors stated that these negative markers provide a list of potential targets for therapies. However, the results focus more on their prognostic potential. Authors need to either expand on the therapeutic potential by mentioning ongoing clinical trials or revise the statement.

We have removed the simple summary as this is not required for IJMS.  We have revised the introduction to focus more on the prognostic potential.

“We also identified TGFB2-independent marker genes showing a negative correlation between mRNA expression and OS across all mRNA expression levels and validated with an independent data set from the KMplotter database. The increased HR for patients with high levels of expression of these marker genes suggests that these marker genes could be presented as future development for targeted therapies.” (Lines 123-128)

“A multivariate analysis revealed that when controlling TGFB2 expression, we identified six genes (ENO1, GLRX2, PLOD1, PRDX4, TAGLN2, and TMED9) that exhibited a negative correlation between mRNA expression and OS across all expression levels in the breast cancer patient, validated with an independent data set from the KMplotter database.  Patients expressing low levels of these 6 genes would benefit from standard therapies as they showed improved survival outcomes at low mRNA levels of expression. Five of these negative prognostic markers are druggable (ENO1, PLOD1, PRDX4, TAGLN2, and TMED9). The increased HR for patients with high levels of expression of these marker genes suggests that these five genes could be presented as future development for targeted therapies.” (Lines 887-896)

Comment 2

In the introduction, the paragraph beginning on line 71 appears unrelated. It is recommended to move it to the discussion if relevant, which will help make the introduction more concise.

We have revised the manuscript to remove the paragraph

Comment 3

The last paragraph of the introduction repeats information from other sections and can be summarized more concisely.

We have now revised the last paragraph of the introduction.

“We utilized a bioinformatic-driven approach to characterize the impact of TGFB2 mRNA, in combination with potential prognostic markers, on overall survival in breast cancer patients from The Cancer Genome Atlas (TCGA) database. We implemented a multivariate Cox proportional hazards model to directly compare hazard ratio (HR) calculations for TGFB2 mRNA, a marker gene expression, including an interaction term of TGFB2 by marker gene expression, while controlling for age at diagnosis, breast cancer subtypes, and comparing patients who received chemotherapy-only therapies, and filtered genes that reported HR greater than 1 in the multivariate model. The interaction term in the model enabled the identification of TGFB2/marker gene combinations that result in synergistic improvements for breast cancer patients. This resulted in TGFB2-dependent positive prognostic markers that could be potentially used as inclusion criteria in biomarker-guided clinical trial designs. We also identified TGFB2-independent marker genes showing a negative correlation between mRNA expression and OS across all mRNA expression levels and validated with an independent data set from the KMplotter database. The increased HR for patients with high levels of expression of these marker genes suggests that these marker genes could be presented as future development for targeted therapies.” (112-128)

Comment 4

AI-Powered PubMed Search is innovative but underdeveloped. The Oncotelic Chatbot is referenced, yet no details on its architecture, query refinement, or bias mitigation are provided. Authors need to provide more details in the method section or supplementary materials.

We thank the reviewer for this suggestion, and we have now provided a detailed supplementary methods section that addresses the use of both the Oncotelic chatbot and Perplexity AI to gather as many relevant references as possible to aid in the writing of the introduction and discussion sections.

Comment 5

Authors are advised to include some of the data presented in supplementary tables as main figures or as a table in the main text, particularly the HR of selected genes.

We thank the review for this suggestion to improve readability.  We have added tables to figures 1, 2 and 6, and included a new table 1 that illustrate the statistics, so to reduce the reporting in the results text.

Comment 6

The quality of Figure 2 indeed needs to be improved.

We have now included a high-resolution tiff figure

Comment 7

Authors are advised to mention the HR of genes only once, as they were repeated multiple times.

We have revised the manuscript according to the suggestion.

Comment 8

Add P values for Figure 4

We have revised the manuscript according to the suggestion.

Comment 9

To elevate significance, discuss how these markers inform specific therapies such as TGFB2 inhibitors or integrate with current standards like PAM50/Oncotype DX.

We thank the reviewer for this suggestion.  We have performed STRING analyses to illustrate the TGFB2 interaction with these 2 breast cancer gene signatures. We have added 2 new figures (4 and 5), and the following discussion:

“Since our multivariate prognostic model identified TGFB2-dependent prognostic markers across the TCGA cohorts including Unclassified, Basal, Normal, Luminal A, Luminal B, and HER2 positive we performed STRING network analysis on the PAM50 gene signature that classifies cancers according to  Luminal A, Luminal B, HER2-enriched, Basal-like and Normal-Like and has been further developed to by combining risk scores to improve future recurrence risk [67,68]. Our analysis revealed that TGFB2 was linked to the network of the PAM50 gene signature through MYC and EGFR. In cancer cell lines, studies have demonstrated that genetic or pharmacological inhibition of MYC in MCF10A basal breast cells results in increased sensitivity to TGFβ-stimulated invasion and metastasis [69], and signaling via Smad3 and ERK/Sp1 mediate TGF‐β‐induced EGFR upregulation [70]. This suggests that TGFB2 mRNA expression levels in breast cancer tumors may be used in conjunction with the PAM50 gene signature to assess the prognostic impact on patients.  In particular, the mRNA expression of GPC4, SULF1, ANTXR1, ITGA11, and COL10A1 from our multivariate Cox proportional hazards models exhibited a significant statistical interaction with TGFB2 mRNA. TGFB2 was a positive prognostic factor, and the five genes were negative prognostic factors from the HR calculations of multivariate models (Table S1). Our results suggest that combinations of high levels of TGFB2 mRNA and low levels of the 5 marker gene expression will improve patient OS outcomes, and this can be used in conjunction with the PAM50 gene signature.

OncotypeDx is a genomic test used for early-stage, hormone receptor-positive, HER2-negative breast cancer to help guide treatment decisions, specifically regarding chemotherapy. It analyzes the expression of 21 genes in a tumor sample to provide a Recurrence Score from 0 to 100. This score indicates the risk of cancer returning and the potential benefit from chemotherapy [71]. STRING network analysis showed associations of this gene signature with TGFB2 via TGFB1 associations with CD68, PGR, ESR1, BCL2 and ERBB2. TGF-β1, ERBB2, and CD68 forms a signaling network that impacts cancer progression, invasion and resistance to therapy in HER2-overexpressing breast cancer tumors [72,73]. There was no direct association of TGFB2 mRNA and the OncotypeDX gene signature.” (Lines 535-562)

Comment 10

7 positive markers mentioned in lines 19–20 conflict with the 11 markers in lines 248–249; please clarify.

We have now clarified how the 2 sets of markers were screened:

“We employed two screens to identify TGFB2-dependent markers: 1. In our first screen the expression of genes that exhibited significant prognostic impact for TGFB2 mRNA expression (p < 0.05) AND Gene 2 expression (p < 0.05) AND TGFB2 by Gene2 interaction terms (p<0.05); 2. The second screen considered marker genes (Gene2), which showed significant improvement in OS outcomes when comparing TGFB2high/Gene2high versus TGFB2high/Gene2high-expressing cohorts of patients using Kaplan-Meier analysis.

We identified negative prognostic markers for the whole cohort of breast cancer patients using Kaplan-Meier analysis of the marker Gene2 that exhibited worse OS times at high levels of Gene2 expression in both TCGA and KMplotter datasets.” (Lines 849-857)

“In our first screen, we sought to identify TGFB2-dependent marker genes that exhibited significant HR calculations for TGFB2 mRNA expression (p < 0.05) AND Gene 2 expression (p < 0.05) AND TGFB2 by Gene 2 interaction terms (p<0.05) (Figure 1; 11 marker genes upregulated in tumors). Expression of COL10A1 mRNA exhibited a significant (p < 0.0001) 877.9-fold change in tumor compared to normal tissue; this high value for fold-change was a result of the very low expression of COL10A1 in normal tissue (Mean ± SEM = −3.92 ± 0.16 log2TPM). Three genes showed a greater than 3-fold increase in expression in tumors compared to normal tissue: ITGA11 (3.88-fold), SULF1 (5.19-fold), and HSD17B6 (8.34-fold), all with p < 0.0001 (Figure 1B).

Examination of the multivariate Cox proportional hazards model revealed two sets of marker genes from the first screen (Table S1). Six of these genes showed an increase in hazard ratios (HR) even when considering the positive prognostic impact of TGFB2 mRNA and significant interaction terms in the Cox model (Table S1): HSD17B6; ITGA11; GPC4; COL10A1; ANTXR1; and SULF1 (HR ranged from 1.43 to 1.22). Five genes showed a decrease in HR of both TGFB2 and Gene2 using the multivariate Cox proportional hazards model that is multiplied as suggested by the significant TGFB2 by Gene2 statistical interaction effects: ARMC7; TMEM14B; AMFR; AFMID; and GDAP1 (HR ranged from 0.02 to 0.62) (Table S1).” (Lines 140-148)

“We next focused of marker genes that exhibited positive prognostic impact in combination with TGFB2 mRNA expression from the list of 380 marker genes (Gene2)  that showed significant interaction parameter from the Multivariate Cox proportional hazards models.  Seven marker genes GDAP1, TBL1XR1, RNFT1, HACL1, SLC27A2, NLE1, and TXNDC16 showed significant improvement in OS outcomes comparing TGFB2high/Gene2high versus TGFB2high/Gene2high expressing cohorts of patients using Kaplan-Meier analysis; shown by separation for two arms of the four survival curves (p < 0.05), whereby high levels of TGFB2 and Gene2 mRNA expression resulted in the most favorable survival outcomes (Figure 2). Calculation of the multivariate parameters for these 7 marker genes reported p-values for the interaction term ranging from 0.004 to 0.036 (Table S2). Four of these genes were significantly upregulated in tumor tissues (p < 0.0001 for all comparisons) (Figure 2H); SLC27A2; TXNDC16; TBL1XR1; and GDAP1.  Kaplan-Meier analysis of patient cohorts with above median expression of TGFB2 and GDAP1 mRNA (n = 181; p = 0.0084, Figure 2A); TBL1XR1 mRNA (n = 222; p = 0.0121, Figure 2B); RNFT1 mRNA (n = 190; p = 0.0164, Figure 2C); HACL1 mRNA (n = 184; p = 0.0188, Figure 2D); SLC27A2 mRNA (n = 177; p = 0.03, Figure 2E); NLE1 mRNA (n = 196; p = 0.031, Figure 2F); and, TXNDC16 mRNA (n = 185; p = 0.036, Figure 2G) showed significantly improved survival outcomes than patients cohorts expressing high mRNA levels of TGFB2 and low levels of the marker genes (Figure 2, Table S3).” (Lines 171-189)

Comment 11

COL10A1's 877.9-fold change seems extreme; verify log2TPM calculations. A figure showing the full heatmap would aid interpretation.

We have provided the full heatmap as a supplemental figure. This high fold change was a result of very low level of expression in normal tissue of less than 1 trancript per 10 million in normal tissue.  We checked the log2 calculations.  We have revised the statement on line 146:

“Expression of COL10A1 mRNA exhibited a significant (p < 0.0001) 877.9-fold change in tumor compared to normal tissue; this high value for fold-change was a result of the very low expression of COL10A1 in normal tissue (Mean ± SEM = −3.92 ± 0.16 log2TPM).”

Comment 12

Authors need to refine the discussion and avoid repeating results data in this section, as well as avoid mentioning tables and figures.

We have revised the manuscript according to the suggestion.

Comment 13

Authors are requested to expand the discussion to address limitations.

We have expanded the limitations as requested:

“The present study identified biomarkers using a bioinformatics-led approach, without the need for additional laboratory testing of marker mRNA and protein levels. Our study identified biomarkers that were positively prognostic at high levels of TGFB2 mRNA expression, suggesting that these biomarkers were highly context-dependent.  Protein-protein association network maps did not show a direct mechanism of TGFB2 action. Future analysis will require cell-level expression in the tumor microenvironment to determine the localization of TGFB2, TGFB receptors, and biomarkers, thereby elucidating the establishment of TGFB2 dependency. Thus, a validation pipeline combining qRT-PCR, immunohistochemistry (IHC), and single-cell RNA sequencing (scRNA-seq) will enable comprehensive confirmation of biomarkers by integrating bulk quantitative mRNA measurement, spatial protein localization, and single-cell resolution of tumor heterogeneity. Initial qRT-PCR on bulk tumor RNA quantifies marker mRNA levels and validates differential expression, followed by IHC to analyze protein expression patterns and distribution in formalin-fixed tissues. Concurrently, scRNA-seq of fresh tumor samples will reveal cell–type–specific expression, heterogeneity, and involvement in tumor microenvironment dynamics. Integrating these complementary approaches ensures robust validation by confirming molecular signatures predicted by bioinformatics, correlating mRNA and protein data, and elucidating the roles of biomarkers in distinct cellular subpopulations, thereby accelerating clinical translation and personalized therapy development.” (Lines 692-711)

Comment 14

Authors are requested to explain how these markers might stratify patients for anti-TGFB2 trials.

The prognostic analysis showed that high levels of TGFB2 results in a favorable OS outcome in combination with the marker genes, therefore we cannot propose to use anti-TGFB2 therapies in breast cancers.

Comment 14.

Authors are requested to mention Future directions, such as scRNA-seq integration.

See comment 13.  We included limitations and future studies in this section.

Comment 15

Some authors with a conflict of interest are involved in designing the study. The particular role of the author needs more details.

SQ is an independent consultant for Oncotelic Therapeutics and has no shares or any other financial interests in the company.  The consultancy requires a rigorous appraisal of the potential use of anti-TGFB2 therapies.  Indeed, the bioinformatic analysis carried out in this present study does not warrant the use of anti-TGFB2 therapies in breast cancers. Expression of TGFB2 mRNA could be used as a positive prognostic biomarker in combination of the identified marker genes to include patients susceptible to standard therapies.  This finding provides balance to other published papers by Oncotelic, which suggest the use of anti-TGFB2 therapies for patients with high levels of TGFB2 mRNA in cancers such as gliomas and pancreatic cancers.  The TGFB2 drug candidate is still going through clinical development- so there is no commercial conflict of interests.  The bioinformatic efforts are independent and hypothesis-generating and point out potential research opportunities, just like efforts at academic laboratories.

Comment 16

There are some typos and grammatical errors. Authors are encouraged to revise the manuscript grammatically.

We have revised the manuscript as requested.

Comment 17

Authors need to double-check their manuscript for plagiarism and avoid using parts of other published papers, particularly in section 2.2 and Lin 89-99

We have revised the section to read:

“We analysed gene expression by downloading log2-transformed transcripts per million (TPM) RNAseq summary files (see https://toil-xena-hub.s3.us-east-1.amazonaws.com/download/TcgaTargetGtex_rsem_gene_tpm.gz for full metadata) from the UCSC Xena web platform (https://xenabrowser.net/datapages/, accessed 25/07/2023)  [20]. This enabled a comparative study of 179 normal breast tissue samples (“GTEX Breast”) and 786 breast cancer patient samples with associated clinical data (“TCGA Breast Invasive Carcinoma”). The data are sourced from the UCSC Toil RNAseq recompute compendium, which provides a harmonised dataset with realigned and recalculated gene and transcript expression levels for all TCGA, TARGET, and GTEx samples [21], facilitating direct comparisons of gene expression between TCGA tumour samples and matched GTEx normal tissues.” (Lines730-740)

“We applied a two-way hierarchical clustering approach to arrange gene expression profiles, grouping together both samples and genes with comparable mRNA expression patterns. This was achieved by employing the average linkage method and the default Euclidean distance metric, as implemented via the heatmap.2 function from the R package gplots_3.1.1. The resulting cluster visualization illustrated the mean expression in tumor samples, normalized to the average expression observed in normal breast cancer tissue, and presented as log2-transformed fold-change values. Dendrograms were generated for both the rows (genes) and the columns (samples), thereby displaying the organization and relationship of co-expressed genes across the patient cohort.” (Lines 770-778)

Round 2

Reviewer 1 Report

Comments and Suggestions for Authors

The authors have provided clarifications and explanations to the comments provided upon review of the manuscript. Especially, the supplementary file is helpful. Their responses to the reviewer comments seem justifiable. Therefore, I accept the paper for publication with editor's approval

Author Response

The authors have provided clarifications and explanations to the comments provided upon review of the manuscript. Especially, the supplementary file is helpful. Their responses to the reviewer comments seem justifiable. Therefore, I accept the paper for publication with editor's approval

We thank the reviewer for the positive response to our revisions. We feel the manuscript has improved in clarity.

Reviewer 4 Report

Comments and Suggestions for Authors

All gene names should be italicized throughout the manuscript, including in the main text and all figure panels.

The manuscript contains multiple inconsistencies and typographical errors in abbreviations. Please review all abbreviations carefully and ensure that each term is written in full at first mention, followed by the appropriate abbreviation. Examples requiring revision include: quantitative real-time polymerase chain reaction (line 407), immunohistochemistry (line 419), and GDAP1 (line 500).

The findings and conclusions presented in this study rely on correlations derived from gene-protein, gene-gene, and protein-protein interaction data obtained from biological databases. While such relationships may reflect meaningful associations, they do not confirm causal or mechanistic links without experimental validation. Therefore, the statements in the Abstract and Conclusion should be carefully revised to avoid overstated interpretations.

Even if interactions among genes or proteins show strong correlations, these do not inherently imply biological causality. For example, claims such as: “This finding implies these individuals could derive greater benefit from treatments for patients expressing high levels of TGFB2 and the marker genes”, “STRING analysis revealed that TGFB2 is linked to EGFR and MYC from the PAM50 breast cancer gene signature, indicating TGFB2-related markers may help assess prognosis in breast cancer patients alongside PAM50” or “The worst survival outcomes and increased HR for patients with high expression levels of these genes suggest that these five genes could be proposed as future therapeutic targets, or that patients with low expression should be excluded from clinical trials” should be avoided, as they imply causation and direct clinical applicability without supporting experimental evidence. Rather than presenting these findings as predictive or therapeutic conclusions, the authors should discuss them as correlation-based observations and frame their interpretations accordingly, using cautious and scientifically appropriate language.

Author Response

  1. All gene names should be italicized throughout the manuscript, including in the main text and all figure panels.

We have now italicized all gene names when referring to genes and mRNA in the text and the figures/tables.

  1. The manuscript contains multiple inconsistencies and typographical errors in abbreviations. Please review all abbreviations carefully and ensure that each term is written in full at first mention, followed by the appropriate abbreviation. Examples requiring revision include: quantitative real-time polymerase chain reaction (line 407), immunohistochemistry (line 419), and GDAP1 (line 500).

We thank the reviewer for their careful reading of the manuscript and for pointing out the inconsistencies.  The full gene names are provided at first mention, and the abbreviated gene names are used thereafter.  In addition to abbreviating the experimental techniques mentioned as examples, we have found additional terms to abbreviate, eg, scRNA-seq (Line 398), chemo-only (Line 141), TPM (Line 151), DFS (Line 356), EMT (Line 379), HCC (Line 380).

“Expression of collagen type x alpha 1 chain (COL10A1) mRNA exhibited a significant (p < 0.0001) 877.9-fold change in tumor compared to normal tissue; this high value for fold-change was a result of the very low expression of COL10A1 in normal tissue (Mean ± SEM = −3.92 ± 0.16 log2TPM). Three genes showed a greater than 3-fold increase in expression in tumors compared to normal tissue: integrin subunit alpha 11 (ITGA11), sulfatase 1 (SULF1), and hydroxysteroid 17-beta dehydrogenase 6 (HSD17B6), all with p < 0.0001 (Figure 1B).

Examination of the multivariate Cox proportional hazards model revealed two sets of Gene2 markers from the first screen (Table S1). Six of these genes showed an increase in HR even when considering the positive prognostic impact of TGFB2 mRNA and significant interaction terms in the Cox model (Table S1): HSD17B6; ITGA11; glypican 4 (GPC4); COL10A1; ANTXR cell adhesion molecule 1 (ANTXR1); and SULF1 (HR ranged from 1.43 to 1.22). Five genes showed a decrease in HR of both TGFB2 and Gene2 using the multivariate Cox proportional hazards model that is multiplied as suggested by the significant TGFB2 by Gene2 statistical interaction effects: armadillo repeat containing 7 (ARMC7); transmembrane protein 14B (TMEM14B); autocrine motility factor receptor (AMFR); arylformamidase (AFMID); and ganglioside induced differentiation associated protein 1 (GDAP1) with HR ranging from 0.02 to 0.62 (Table S1).” (Lines 148-165)

“Seven Gene2s: GDAP1, TBL1X/Y Related 1 (TBL1XR1), ring finger protein, transmembrane 1 (RNFT1), 2-hydroxyacyl-CoA lyase 1 (HACL1), solute carrier family 27 member 2 (SLC27A2), notchless homolog 1 (NLE1), and thioredoxin domain containing 16 (TXNDC16)” (Lines182-185)

  1. The findings and conclusions presented in this study rely on correlations derived from gene-protein, gene-gene, and protein-protein interaction data obtained from biological databases. While such relationships may reflect meaningful associations, they do not confirm causal or mechanistic links without experimental validation. Therefore, the statements in the Abstract and Conclusion should be carefully revised to avoid overstated interpretations.

We thank the reviewer for encouraging a more careful examination of the interpretations of our results and we have revised the abstract and conclusions accordingly to reflect the hypothesis-generating and correlative nature of the findings. We have removed “predictive” from the title to clarify that these markers are prognostic only.

“We conducted a hypothesis-generating study using a bioinformatics approach in order to identify potential prognostic biomarkers for breast cancer patients across multiple molecular subtypes.” (Lines 14-16)

“In cases dependent on TGFB2, increased mRNA expression of TGFB2 alongside higher levels of GDAP1, TBL1XR1, RNFT1, HACL1, SLC27A2, NLE1, or TXNDC16 was correlated with improved OS among breast cancer patients, of which four genes were upregulated in tumor tissues (SLC27A2, TXNDC16, TBL1XR1, GDAP1). Future studies will be required to confirm breast cancer patients could improve OS outcomes for patients expressing high levels of TGFB2 and the marker genes in prospective clinical trials. Additionally, multivariate analysis revealed that the elevated expression of six genes (ENO1, GLRX2, PLOD1, PRDX4, TAGLN2, TMED9) were correlated with increases in HR, independent of TGFB2 mRNA expression; all except GLRX2 were identified as druggable targets. Future investigations assessing protein expression in breast cancer tumors to confirm the results of our retrospective analysis of mRNA levels will determine whether the protein products of these genes represent viable therapeutic targets. Protein-protein interaction (STRING) analysis indicated that TGFB2 is associated with EGFR and MYC from the PAM50 breast cancer gene signature. These findings suggest that correlation of TGFB2-related markers could potentially complement the PAM50 signature in the assessment of OS prognosis in breast cancer patients, but further validation of the TGFB2/EGFR/MYC proteins in tumors is warranted.” (Lines  28-44)

TGFB2-dependent biomarkers, elevated TGFB2 mRNA expression is a prognostic biomarker associated with breast cancer patients correlated with improved OS outcomes at high expression levels of GDAP1, TBL1XR1, RNFT1, HACL1, SLC27A2, NLE1, and TXNDC16. This suggests that patients with high levels of TGFB2 and Gene2 were correlated with improved OS and used as markers for future prospective clinical trials.  A multivariate analysis revealed that when controlling TGFB2 expression, we identified six genes (ENO1, GLRX2, PLOD1, PRDX4, TAGLN2, and TMED9) that exhibited a negative correlation between mRNA expression and OS across all expression levels in the breast cancer patient, validated with an independent data set from the KMplotter database.  Five of these negative prognostic markers are druggable (ENO1, PLOD1, PRDX4, TAGLN2, and TMED9). The increased HR for patients with high levels of expression of these Gene2 suggests that these five genes could be presented for targeted therapies following confirmation of protein levels and drug action in breast cancer patients. Examination of the protein-protein interaction networks suggested that the correlation of TGFB2-related markers could potentially complement the PAM50 signature in the assessment of OS prognosis in breast cancer patients following validation of the TGFB2/EGFR/MYC protein associations in tumors.” (Lines 878-894)

  1. Even if interactions among genes or proteins show strong correlations, these do not inherently imply biological causality. For example, claims such as: “This finding implies these individuals could derive greater benefit from treatments for patients expressing high levels of TGFB2 and the marker genes”, “STRING analysis revealed that TGFB2 is linked to EGFR and MYC from the PAM50 breast cancer gene signature, indicating TGFB2-related markers may help assess prognosis in breast cancer patients alongside PAM50”

We have now revised the statements to highlight the correlative nature of the findings.

“It is noted that the TGFB2 association represent a strong correlation, future controlled experiments in reduced systems will be required to infer mechanistic insight of these TGFB2-association in breast cancer tissues” (Lines 531-533)

“Our results suggest that combinations of high levels of TGFB2 mRNA and low levels of these five genes resulted in improving patient OS outcomes. This correlation suggests that TGFB2 mRNA expression levels in breast cancer tumors may be used in conjunction with the PAM50 gene signature to assess the prognostic impact on patients following validation in prospective clinical trials. “ (Lines 547-551)

or

  1. The worst survival outcomes and increased HR for patients with high expression levels of these genes suggest that these five genes could be proposed as future therapeutic targets, or that patients with low expression should be excluded from clinical trials”

We have now revised claims regarding how our results can be interpreted.

“The network analysis showed no direct mechanistic connection of TGFB2 and SULF1, but they are components of a highly interconnected set of associations. Further work will need to elucidate the expression of TGFB1, TGFB2, and SULF1 using scRNA-seq studies for the distribution of expression in the cellular compartments in tumors.” (Lines  423-427)

“Therefore, patients expressing high mRNA levels of these two genes displayed improved OS outcomes to standard therapies that requires confirmation in prospective clinical trials.” (Lines 470-472).

“Our results suggest that ENO1 is targetable for treated patients with high levels of ENO1 mRNA. The prognostic impact at the mRNA level will need to be confirmed by measuring protein levels in the tumors.” (Lines 594-596).

“Our results suggest that breast cancer patients with low levels of GLXR2 expression exhibited improved OS outcomes with standard therapies. Prospective clinical trials examining GLXR2 mRNA levels will be required to validate our observations.” (Lines 608-611)

“Our study suggests that patients expressing high levels of PLOD1 can be targeted for therapies against Prolyl 4-hydroxylase alpha subunits pending confirmation that increase in PLOD1 mRNA results in increases in PLOD1 protein levels in breast cancer patients.” (Lines 627-630).

“Our correlative results suggest that breast cancer patients with high levels of PRDX4 may be targeted using compounds such as Con A pending confirmation of increased protein levels are also prognostic in breast cancer patients.” (Lines 648-650).

“Although no clinical trials are testing these compounds, inhibition of TAGLN2 may be pursued as a therapeutic option for breast cancer patients expressing high levels of this gene following confirmation of the prognostic impact of the protein in breast cancer patients.” (Lines 676-678).

“Our results suggest that breast cancer patients with high levels of TMED mRNA are potentially susceptible to inhibiting TMED pending confirmation of protein levels.” (Lines 685-687).

should be avoided, as they imply causation and direct clinical applicability without supporting experimental evidence. Rather than presenting these findings as predictive or therapeutic conclusions, the authors should discuss them as correlation-based observations and frame their interpretations, accordingly, using cautious and scientifically appropriate language.